# Ucenprubart is an agonistic antibody to CD200R with the potential to treat inflammatory skin disease: preclinical development and a phase 1 clinical study

Anja Koester [1] ✉, Derrick R. Witcher[2], Mark Lee[3], Stephen J. Demarest[4,6], Scott Potter[1], Katie Werle[1,7], Scott Bauer[2], Diana Ruiz[4], Laurent Malherbe[5], Josh Poorbaugh[2], Andrew Glasebrook[1,8], Christoph Preuss [2], Gourab Datta[2], Ziqiao Wang[2,9], Jack Knorr[2], David Manner[2], Dipak Patel[1], Carsten Schmitz[1], Paul Klekotka[1] & Ajay Nirula[1,10]

CD200R is a checkpoint inhibitory receptor central to the pathogenesis of inflammatory skin disease. Here we describe the development and phase 1 clinical study (NCT03750643) of ucenprubart, a CD200R agonist antibody to downregulate immune system inflammation. Preclinical studies find ucenprubart inhibiting Fcγ receptor-induced cytokine secretion from myeloid cells in vitro and demonstrating efficacy in a mouse contact hypersensitivity model. The randomized, placebo-controlled, NCT03750643 trial assesses safety and pharmacokinetics in healthy subjects, and efficacy in atopic dermatitis patients. The primary efficacy outcome is the proportion of patients achieving Validated Investigator's Global Assessment for Atopic Dermatitis (vIGA-AD) 0 or 1 with ≥2-point improvement from baseline at week 12. Secondary outcomes are proportions of patients achieving the primary outcome and mean changes in Eczema Area and Severity Index (EASI) and SCORing Atopic Dermatitis (SCORAD) across weeks 1 through 12, and cutoffs at week 12. Sixty-two healthy participants and 40 patients are enrolled. No serious adverse events or discontinuations due to adverse events is seen with ucenprubart. The primary endpoint is not met; however, overall improvements are observed in EASI-75 and SCORAD through 12 weeks. CD200R may be a promising therapeutic target for treating autoimmune disease, including inflammatory skin diseases.

CD200R is a checkpoint inhibitory receptor primarily expressed on the surface of innate immune system cells, specifically those of the monocytic lineage like macrophages, mast cells, and dendritic cells, as well as on activated T-cell subsets such as memory T cells[1,2]. The natural ligand for CD200R is CD200, which is more broadly expressed on multiple cell types including lymphocytes and non-hematopoietic cells. The extracellular regions of CD200 and CD200R contain 2 highly glycosylated immunoglobulin (Ig) superfamily domains followed by a single-pass transmembrane domain. Upon interaction of CD200R and CD200 via their extracellular Ig-like domains, tyrosine motifs in the

**Fig. 1 | CD200R system and in vitro binding and activity profiles of ucenprubart. a** CD200R and CD200 system; the receptor and ligand belong to the IgG superfamily of proteins. Binding of CD200 to CD200R initiates an intracellular signaling cascade mediated by the recruitment of adapter protein Dok1/2 and subsequent downmodulation of pERK. CD200R is a member of the paired-receptor family, where a homologous activating form (CD200RLa) exists with opposite activity due to coupling to different intracellular signaling molecules (not shown). **b** Flow cytometry showed ucenprubart binding to human (huCD200R) and cynomolgus (cyCD200R) CD200R but not to cyCD200RLa in CHO cells recombinantly expressing the respective receptors. Note that ucenprubart has higher affinity to cynomolgus (green triangles) versus human (blue squares) receptor (Supplementary Table 1); as a result, the binding signal is higher for cynomolgus receptor-expressing cells. No binding was detected on cells expressing cyCD200RLa and non-transfected cells (downward red triangles and black circles, respectively, overlapping along the *x*-axis). Values are the average of two technical repeats. **c** A co-binding experiment using flow cytometry showed CD200 ligand (hCD200-Fc) bound to CD200R when cells were preincubated with ucenprubart, hCD200Fc, or isotype control antibody at 66 nM. Values are the average of two technical repeats. **d** In ovalbumin-activated T cells co-cultured with CD200-expressing B cells, IL-2 release was inhibited by ucenprubart but was increased by an anti-CD200 antibody. Values shown are compiled from multiple experiments and normalized to the percentage of the baseline for comparison. Actual IL-2 values after stimulation with ovalbumin peptide varied from 800 to 1200 pg/mL compared to <5 pg/mL unstimulated. CHO Chinese hamster ovary, ctrl control, Dok1/2 downstream of tyrosine kinases 1 and 2, Fc crystallizable fragment, IgG immunoglobulin G, IL interleukin, mAb monoclonal antibody, MFI mean fluorescence intensity, RasGAP RAS p21 protein activator 1, SEM standard error of the mean. Source data is provided as a Source Data file.

intracellular domain of CD200R are phosphorylated and signal via recruitment of downstream of tyrosine kinases 1 and 2 (Dok1/2) (Fig. 1a).

In vivo findings indicate that deletion of the *CD200R* or *CD200* genes in mice results in no overt phenotype but increases susceptibility to the induction of autoimmune diseases[3]. Conversely, overexpression of CD200 in mice provides resistance to allogeneic transplantation rejection and dextran sulfate sodium-induced colitis[3,4]. Increased expression of CD200 has also been associated with poor prognosis in several liquid and solid tumors[5,6], and the CD200R-CD200 pathway is a known regulator of the tumor microenvironment[7–10]. Importantly, the *CD200* gene has been mimicked by some viruses as an immune escape mechanism, further underlining the immunosuppressive function of CD200R[11,12]. Recently, the receptor was described as identifying cells of the Th2 response in allergic diseases[13]. These cell types contribute to the pathology of diseases such as atopic dermatitis (AD); thus, CD200R agonist antibodies could potentially attenuate the activity of these cells in AD. Several independent large-scale case–control studies have identified associations between CD200R and/or its downstream signaling molecule Dok2 and eczema and allergic disease, further supporting their role in AD[14–16]. In addition, fine mapping and expression quantitative trait loci analysis revealed that a common missense variant in CD200R (*rs9865242, p.E312Q*), which lowers *CD200R1* transcript

levels, is predicted to increase disease susceptibility by disrupting Dok2-mediated downstream signaling[14]. Efforts to augment CD200R inhibitory function using a recombinant version of the ligand, CD200 fragment crystallizable (Fc)[17], or an antibody[18] in mice have demonstrated that activating CD200R suppresses disease activity in experimental multiple sclerosis and autoimmune-mediated uveoretinitis models. Increasing CD200R-mediated signaling constitutes a potential approach to manage monocyte- and Th2-mediated autoimmune disorders, which could lead not only to a short-term impact on disease but potentially also to disease modification and hence durability of response posttreatment[19].

Here we describe the development of the humanized CD200R agonist antibody ucenprubart (LY3454738) as a potential modulator of immune activation in inflammatory disease. Ucenprubart-mediated agonism of CD200R shows inhibition of monocyte cytokine secretion in vitro and improves inflammation in a cutaneous AD disease model in vivo, with a different mechanism than with dupilumab. Ucenprubart improved disease activity measures in a phase 1b study and was safe and well-tolerated over 12 weeks. These findings demonstrate that targeting the immune checkpoint inhibitory receptor CD200R provides opportunities for the treatment of autoimmune or inflammatory diseases such as AD via different mechanisms than those of currently available therapies.

## Results

### Ucenprubart/LY3454738 generation

The objective of the discovery campaign was to identify and develop an agonist antibody to the inhibitory form of both human and cynomolgus CD200R. To that end, we immunized rabbits with alternating soluble extracellular domains (ECDs) of the human and cynomolgus monkey CD200R as antigens. We confirmed binding of the final lead molecule (refer to Methods for screening process) by flow cytometry on cells overexpressing human or cynomolgus CD200R (Fig. 1b). CD200R is a member of the paired receptor family, meaning a closely related homolog of the inhibitory CD200R with opposite activity exists in cynomolgus monkeys (cyCD200RLa). To enable non-human primate toxicological studies, we counter-screened by flow cytometry for non-binding on cells expressing cyCD200RLa; we did not observe binding even at high concentrations (Fig. 1b). We cloned and attempted to express the ECD of human CD200RLa (NP_001008784.2) but failed to yield protein, possibly because human CD200RLa is missing 1 of the 2 canonical cysteines in D1 observed in V-class Ig folds of all other species. This suggests that the human form of CD200RLa may be unstable and not found in humans. We measured the binding affinity of ucenprubart to human and cynomolgus CD200R, as well as to cyCD200RLa and mouse CD200R, using surface plasmon resonance. The binding affinity (dissociation constant $[K_D]$) of ucenprubart is 5.6 nM to human CD200R and 2.3 nM to cynomolgus CD200R, with about 1000-fold lower affinity to cyCD200RLa (2.5 μM) and mouse CD200R (5.1 μM) (Supplementary Table 1).

In addition to avoiding binding to cyCD200RLa, we also sought to target a binding site distinct from the CD200 binding site to avoid disrupting normal interactions between CD200R and CD200. We initially screened all antibodies for binding to CD200R-expressing cells using flow cytometry, followed by a biochemical assay for their ability to bind plate-bound recombinant human CD200R in the presence of the CD200 ligand. We further confirmed the ability of the ligand to bind cell-bound CD200R in the presence of ucenprubart in a flow competition experiment using HEL92.1.7 cells naturally expressing CD200R (Fig. 1c). We incubated cells with ucenprubart, hCD200Fc, or isotype antibody, followed by a titration of the ligand. Binding of the hCD200Fc ligand increased slightly when ucenprubart was prebound to the cells, demonstrating that the antibody does not target the ligand-binding site.

### Ucenprubart augments ligand-mediated receptor activation

To assess agonistic activity under more physiological conditions and in the presence of the CD200 ligand, we used a cell-to-cell interaction assay where the ligand and the receptor are expressed on different cell lines that interact for activation. The interaction of CD200 on A20 cells and CD200R on D011 T cells (permissive for ovalbumin-induced activation) had a mild suppressive effect on ovalbumin-induced activation, which could be eliminated with an anti-CD200 antibody (Fig. 1d). However, ucenprubart further augmented CD200R-mediated inhibition. Thus, ucenprubart can engage and agonize the receptor in the presence of the CD200 ligand and further enhance the inhibitory signal provided through ligand binding.

### Isotype selection for ucenprubart/LY3454738

It is well known that the isotype and hence binding of the antibody Fc-hinge region to Fcγ receptors (FcγRs) can exert a profound impact on antibody function and in vivo efficacy[20]. Based on preclinical mouse model data (not shown), this initially led us to develop an IgG1 isotype of the lead agonistic antibody. Because cytokine release through FcγR activation has been associated with the clinical administration of IgG1 antibodies, we tested the antibody in a human whole blood assay for potential effects on cytokine release. When we incubated the IgG1 isotype variant of LY3454738 in whole blood, interferon (IFN)-γ release was induced for all donors tested (Fig. 2a), suggesting a cytokine release syndrome risk in humans[21]. Therefore, we generated IgG4 variant isotypes and assessed them to identify minimal FcγR binding requirements and maximal efficacy while maintaining safety (ie, avoiding cytokine release). We tested LY3454738 variants with varying degrees of FcγR binding (Supplementary Table 2) in a U937 cell-based assay for functional activity based on suppression of interleukin (IL)-8 release (Fig. 2b). We confirmed the activity in a staphylococcal enterotoxin B-stimulated T-cell proliferation assay where LY3454738 IgG4 S228P (IgG4P) demonstrated slightly reduced activity compared with the IgG1 version (Fig. 2c) but greater activity than that of LY3454738 IgG4P F234A/L235A (IgG4PAA). In addition, when we tested LY3454738 IgG4P in the whole blood cytokine release assay, no cell activation or cytokine release was observed (Fig. 2a), which led us to select LY3454738 IgG4P as the final clinical molecule ucenprubart.

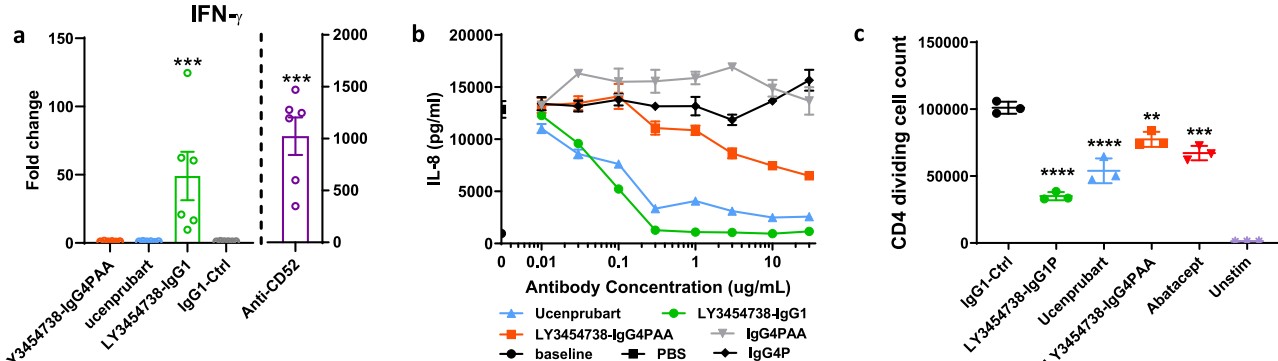

**Fig. 2 | Fcγ receptor binding optimization. a** IFN-γ response after 24-h incubation with 100 μg/ml of LY3454738-IgG4PAA(F234A/L235A), LY3454738-IgG4SP (S228P, ucenprubart), LY3454738-IgG1, control (negative control), mAB1-IgG1 represents isotype control, or anti-CD52-IgG1 (positive control; at 10 μg/ml). Responses are shown for six donors; IFN-γ response is shown as fold change from donor baseline, bars represent the average, and error bars are SEM. *P* values were generated by a paired *t*-test of log$_{10}$-transformed concentrations of treated versus untreated samples; ***$p = 0.001$. **b** Inhibition of IgG1 stimulated IL-8 secretion from U937 cells overexpressing human CD200R. Values are the average of three technical

replicates ±SD. **c** LY3454738 variants inhibited staphylococcal enterotoxin B-stimulated CD4$^+$ T-cell proliferation. Antibodies were added at 30 μg/mL immediately before stimulation for 96 h. The data shown are from a total of three donors. Abatacept (CTLA4-Fc) was used as a positive control for inhibition of T-cell activation. Statistical analysis was done using a paired *t*-test of IgG1 isotype versus treatment groups. **$p = 0.022$, ***$p = 0.001$; ****$p < 0.001$. Ctrl control, Fc crystallizable fragment, hIgG4 human IgG4, IgG immunoglobulin G, IFN interferon, IL interleukin, PBS phosphate-buffered saline. Source data is provided as a Source Data file.

**Suppression of skin inflammation by ucenprubart or dupilumab**

We assessed the immunosuppressive activity of ucenprubart in a humanized mouse model of oxazolone-induced delayed hypersensitivity on mouse ears and compared it to an approved AD agent, dupilumab (anti-IL-4 receptor α [anti-IL-4Rα]). CD34-engrafted NSG mice (huNOG-EXL) were sensitized to the hapten oxazolone and dosed with either ucenprubart or dupilumab 24 h before challenge with hapten on the ear. We determined efficacy by measuring ear thickness before and after the challenge and compared to isotype control antibody-treated animals. The challenge was repeated 3 times. Following the third challenge (day 19 post-sensitization), dupilumab and ucenprubart each reduced swelling by approximately 50% during the 24-h challenge (Fig. 3a).

After we collected cells from mouse ears, assessed quality control, and filtered out cells harboring transcripts that mapped to the mouse genome, the final data set comprised 15,382 human cells (isotype, $n = 4806$; dupilumab, $n = 7703$; ucenprubart, $n = 2873$), with an average of 1799 genes and 5295 transcripts per cell. We visualized the results using uniform manifold approximation and projection (UMAP), which revealed 11 distinct cell clusters that fell into 5 broader cell type categories (Fig. 3b). We annotated cell clusters using distinct transcript markers and a panel of eight CITE-seq (cellular indexing of transcriptomes and epitopes by sequencing) surface markers (see Methods; Supplementary Fig. 1). Our results are in line with a published single-cell study that identified a similar proportion of monocytes, macrophages, basophils, and T cells in the ears of oxazolone-treated mice when compared to untreated controls[22]. CD200R surface protein expression across cells was highest in basophils, macrophages, and dendritic cells, which is consistent with a role in myeloid cell function (Fig. 3c; Supplementary Fig. 1a). We observed no differences in CD200R surface protein expression following the different treatments (Supplementary Fig. 2). When we compared ucenprubart with the isotype, macrophages and monocytes showed the highest number of differentially expressed genes after treatment (Bonferroni-adjusted Wilcoxon rank-sum $p < 0.05$) (Fig. 3d). For dupilumab, differential expression was also found in macrophages and monocytes but was also shown in multiple T-cell types including CD4$^+$ T cells (Fig. 3d). Among the genes affected by dupilumab in CD4$^+$ T cells, we observed a functional enrichment of pathways linked to Th1, Th2, and Th17 response (Supplementary Fig. 3a). While we found that both treatments resulted in a large number of differentially expressed genes in macrophages, only a small number of these affected genes overlapped and were impacted by both treatments (Fig. 3e; Supplementary Data 1). This is consistent with the lack of a correlated treatment response in macrophages as measured by fold-change differences across cell types and treatments (Supplementary Fig. 3b). Gene set and enrichment analysis (GSEA) using curated hallmark gene sets from the Molecular Signature database and KEGG pathway analysis highlighted increased expression of a distinct pathway subset among ucenprubart-treated macrophages compared with the remaining cell types (Fig. 3f, g). The top significantly enriched gene sets among this pathway subset included those associated with general macrophage function and with activation of inflammatory pathways, such as NF-kappa B and JAK-STAT signaling relevant to AD NF-kappa B signaling and JAK-STAT signaling (Fig. 3f, g).

**Phase 1 clinical study**

The promising results from the preclinical development of ucenprubart led to the design of a phase 1 study conducted sequentially in three parts: Part A assessed single-ascending doses in healthy volunteers ($N = 54$); Part B assessed repeat dosing at submaximal exposure in healthy volunteers ($N = 8$); and Part C assessed repeat dosing of ucenprubart at a single dose level (500 mg; selected based on CD200R receptor occupancy characteristics; Supplementary Fig. 4) or placebo in patients with AD ($N = 40$). A CONSORT patient flow diagram is

shown as Fig. 4. A brief summary of pharmacokinetic results for Parts A–C is presented in the Supplementary Results, including the maximum observed drug concentration and the area under the concentration–time curve in Supplementary Table 3.

For patients in Part C randomized to receive placebo ($n = 12$) and ucenprubart ($n = 28$), the mean (standard deviation [SD]) age was 36.8 (15.0) and 42.6 (15.8) years, 50.0% and 60.7% of patients were female, mean (SD) body mass index was 27.7 (4.6) kg/m$^2$ and 28.3 (6.3) kg/m$^2$, and the mean (SD) eczema area and severity index (EASI) total score was 20.5 (9.8) and 20.2 (11.9), respectively (Supplementary Table 4).

Among patients enrolled in Part C, 3 of 12 (25.0%) did not complete treatment with placebo (for reasons of lack of efficacy [$n = 2$] and lost to follow-up [$n = 1$]) and 5 of 28 (17.9%) did not complete treatment with ucenprubart (for reasons of relocation, wanted to try another drug, no reason given, lost to follow-up, and investigator decision [$n = 1$ each]) (Fig. 4).

The proportion of patients who achieved a validated investigator global assessment for atopic dermatitis (vIGA-AD) score of 0 or 1 (0,1) with a ≥2-point improvement from baseline at week 12 was 16.7% ($n = 2/12$) with placebo and 32.1% ($n = 9/28$) with ucenprubart (2-sided Fisher exact test $p = 0.45$) (Fig. 5a). The observed mean percentage change in EASI score from baseline to week 12 was −39.1% with placebo and −60.6% with ucenprubart (Fig. 5b). The proportion of patients who achieved a ≥75% improvement in EASI score (EASI-75) at week 12 was 25.0% ($n = 3/12$) with placebo and 35.7% ($n = 10/28$) with ucenprubart (2-sided Fisher exact test $p = 0.72$) (Fig. 5c). For improvement in itch at week 12, a ≥4-point improvement was seen among a comparable proportion of patients who received placebo ($n = 4/12$ [33.3%]) and ucenprubart ($n = 10/28$ [35.7%]) (Fig. 5d). A statistically significant difference ($p = 0.0157$) was observed in the mean percentage change in EASI score for the combined time points through week 12 between ucenprubart and placebo (Fig. 5b). Similarly, there was a statistically significant difference ($p = 0.004$) in the mean percentage change in the SCORing Atopic Dermatitis (SCORAD) score between ucenprubart and placebo in the combined time points through week 12 (Fig. 5e). Maintenance data after week 12 are shown in Fig. 5f. The mean percentage change in the SCORAD score at week 12 was −36.7% with placebo and −46.4% with ucenprubart (Fig. 5e). For other values at week 12, the number and percentage of patients achieving selected endpoints and the mean percentage change from baseline in EASI score are shown in Fig. 5g.

Given the possibility that a period of dosing with ucenprubart could potentially modulate longer-term immune response, we evaluated the proportion of patients with a continued response through the 12-week follow-up period after dosing (Fig. 5f). Among ucenprubart vIGA-AD responders at week 12, 66.7% ($n = 6/9$) maintained a vIGA-AD response at week 24, while among itch responders at week 12, 70.0% ($n = 7/10$) maintained a ≥4-point improvement in itch at week 24. For EASI-75 responders at week 12, 50.0% ($n = 5/10$) continued to maintain a 75% response in EASI score at week 24. For those with a ≥90% improvement in EASI score at week 12, 57% ($n = 4/7$) maintained a ≥90% response at week 24.

In Part A (single-ascending doses in healthy volunteers), 26 treatment-emergent adverse events (TEAEs) were reported by 19 of 42 (45.2%) healthy participants who received ucenprubart by either intravenous (IV) administration or subcutaneous (SC) infusion, and 4 TEAEs were reported by 2 of 12 (16.7%) healthy participants who received placebo via IV administration. TEAEs were generally of mild intensity, and headache was the most frequently reported TEAE. One injection site reaction, including severe erythema and pruritus of mild severity at the injection site, was reported for 1 participant in Part A from the ucenprubart 100 mg SC cohort.

In Part B (repeat dosing at submaximal exposure in healthy volunteers), 7 TEAEs were reported by 3 of 6 (50.0%) healthy participants who received 2 biweekly doses of ucenprubart 200 mg via IV

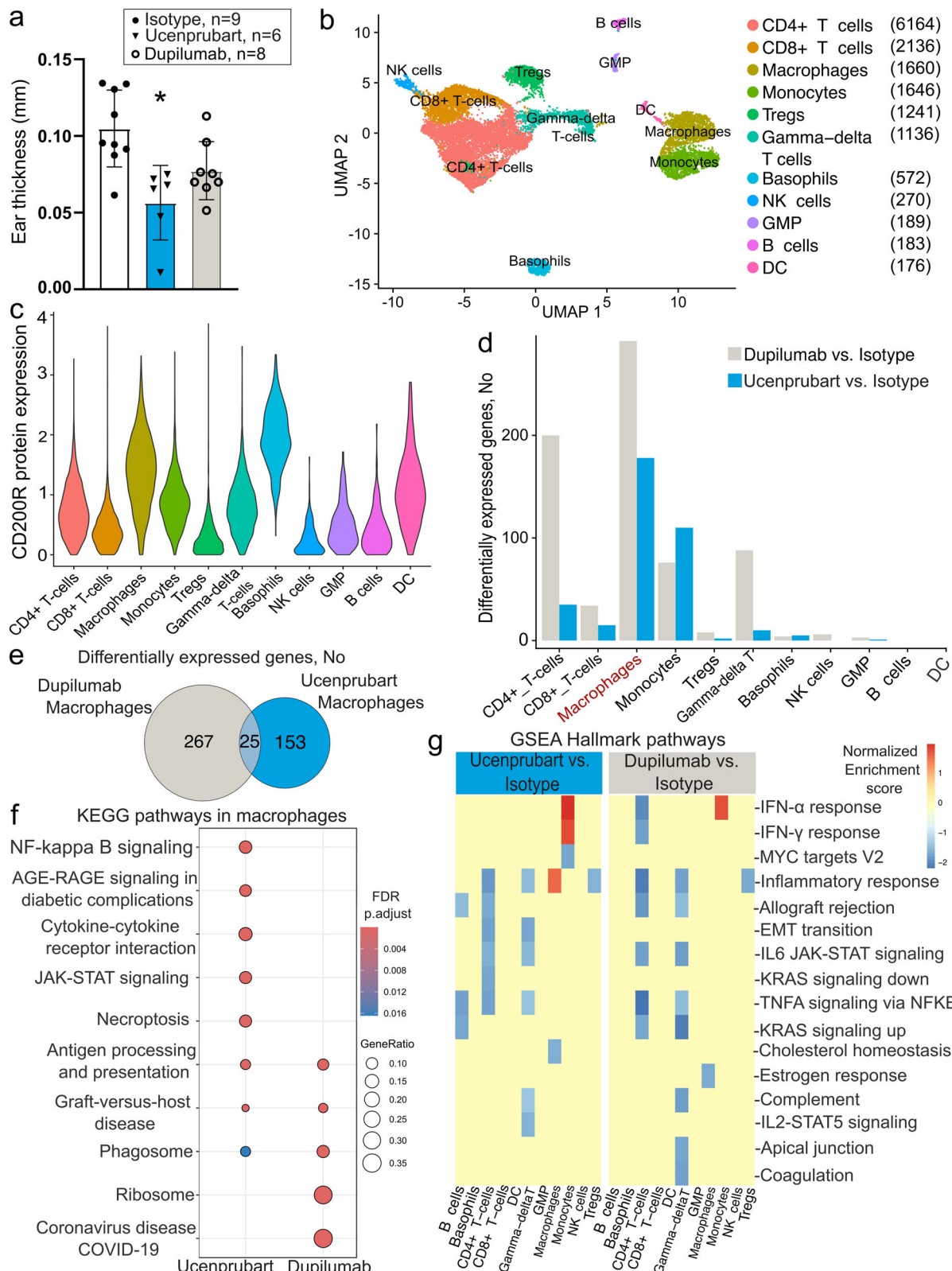

administration. No TEAEs were reported by healthy participants in the placebo arm. All TEAEs with ucenprubart were of mild intensity, and headache was the most frequently reported TEAE.

In Part C, 11 of 28 (39.3%) patients with AD who received ucenprubart reported 25 TEAEs (13 mild and 12 moderate in intensity) (Table 1). Among placebo recipients, 5 of 12 (41.7%) reported 7 TEAEs (6 mild and 1 moderate in intensity). TEAEs reported in >1 patient were

contact dermatitis, urinary tract infection, and contusion, reported in 2 patients each in the ucenprubart treatment arm.

Across the 3-part study, no deaths or serious adverse events (AEs) were reported, no dose-related findings were observed for AEs, and no study treatment discontinuations were reported due to AEs. In addition, there were no reports of infusion-related reactions, abnormalities or clinically meaningful findings in vital signs and

**Fig. 3 | In vivo single-cell-based screening of CD200R. a** Ucenprubart inhibited contact hypersensitivity in humanized mice similarly to dupilumab, based on ear thickness in 23 ears (isotype, $n = 9$; dupilumab, $n = 8$; ucenprubart, $n = 6$) following challenge in oxazolonesensitized mice. *$p = 0.002$ vs isotype using 1-way analysis of variance with the Tukey multiple comparison test. Bars presented as mean values ± standard deviation. **b** Visualization of single-cell RNA sequencing data from human cells derived from humanized NOGEXL mice ears using UMAP. **c** CD200R surface expression, as measured by CITE-seq antibody, was highest in basophils, macrophages, and dendritic cells but did not change with treatment (Supplementary Fig. 2). **d** Differentially expressed genes (familywise error rate-adjusted $p < 0.05$; 2-sided Wilcoxon rank-sum test) highlighted a myeloid cell type-specific response for human cells treated with ucenprubart. Cells treated with dupilumab showed both a myeloid and T-cell-driven response. **e** The overlap between differentially expressed genes across macrophages is depicted as a proportional Venn diagram highlighting distinct effects for both treatments for this cell type. **f** The enriched KEGG (Kyoto Encyclopedia of Genes and Genomes) pathways for differentially expressed genes in macrophages specific to ucenprubart included NF-kappa B signaling and JAK-STAT signaling (Janus kinase (JAK)-signaling transducer of activators of transcription (STAT), which are linked to macrophage function. **g** Gene set enrichment analysis (GSEA) pathway analysis of ucenprubart-treated cells, highlighting a significantly enriched (false discovery rate [FDR]-adjusted $p < 0.05$) subset of pathways from the hallmark Molecular Signature database in macrophages. No. number, CITE-seq cellular indexing of transcriptomes and epitopes by sequencing, DC dendritic cells, GMP granulocyte–monocyte progenitors, Tregs regulatory T cells, UMAP uniform manifold approximation and projection. Source data is provided as a Source Data file.

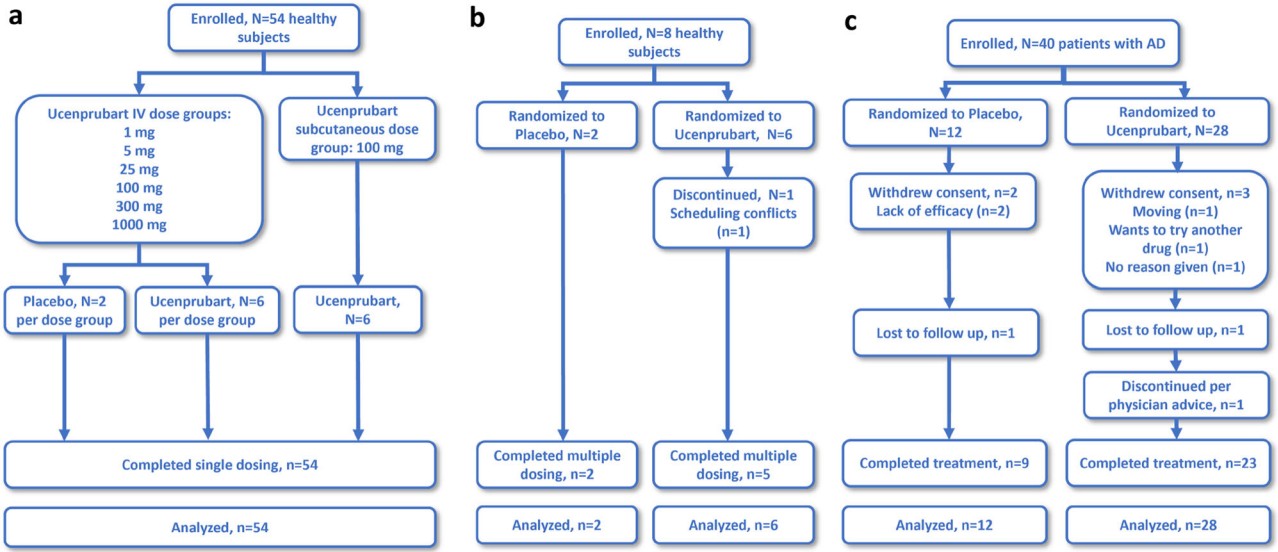

**Fig. 4 | CONSORT diagram for study J1B-MC-FRCC Parts A–C.** CONSORT diagram for **a** healthy subjects in Part A; **b** healthy subjects in Part B; and **c** patients with AD in Part C. AD atopic dermatitis, CONSORT consolidated standards of reporting trials, IV intravenous.

electrocardiogram assessments, or clinically significant alterations or trends in safety laboratory values, including antidrug antibodies.

In the dermis of lesional punch biopsies from patients with AD at baseline and week 12 (Fig. 6), the percentage of cells staining for GATA3 was positively correlated with cells staining for CD3 (Pearson $r = 0.7546$, $p < 0.0001$; Supplementary Fig. 6), suggesting the presence of Th2 cells within this compartment. For CD3, we did not observe within-patient differences between baseline and week 12 lesional samples for either EASI-75 responders or nonresponders (Fig. 6b). For GATA3, 5 of 6 EASI-75 responders showed reduced GATA3 staining in the dermis between baseline and week 12; this trend was not observed for GATA3 among EASI-75 nonresponders (Fig. 6c). When the mean change between baseline and week 12 was calculated, the ucenprubart responder group had a numerically greater reduction in GATA3-positive staining compared with nonresponders (Fig. 6c). CD3 is a general marker of T cells in the skin while GATA3 more specifically identifies Th2 cells in the dermis, suggesting that pathogenic T cells tended to be reduced in ucenprubart responders when compared to nonresponders. In addition, histological examination performed following treatment suggested normalization of the stratum spinosum, basal layer, and stratum corneum in biopsies from week 12 responders, along with increased GATA3 positivity in keratinocytes, consistent with its known role in establishing and maintaining the epidermal barrier[23] (examples in Fig. 6a, hematoxylin and eosin [H&E] and GATA3 staining from the same patient).

## Discussion

Ucenprubart, an agonistic antibody to the human immune checkpoint receptor CD200R, was selected from a candidate pool of monoclonal antibodies (mAbs) and was humanized and engineered for functional activity, binding profile, cytokine release risk, and positive developability attributes. Isotype selection and optimization were required to balance efficacy and safety. CD200R is highly expressed on neutrophils and basophils, and binding of the IgG1 antibody to CD200R on granulocytes can cluster the antibody and induce cytokine release from FcγR-expressing cells, as we observed in the human whole blood cytokine release assay. We found that using an IgG4P isotype, which binds primarily to FcγRI (CD64), induced sufficient clustering of the receptor on the target cells to mediate in vitro and in vivo activity without inducing cytokine release in the human whole blood assay. These results highlight the importance of isotype selection and evaluation of FcγR-dependent IFN-γ release in preclinical cytokine release assays to assess the risk of cytokine release syndrome associated with therapeutic IgG1 antibodies[21]. Another important aspect of our antibody design was to target an epitope away from the ligand-binding site to avoid blocking binding by the endogenous ligand. Lastly, candidate antibodies were counter-screened to avoid reactivity to CD200RLa, an activating form of the receptor found in cynomolgus monkeys and recently described in humans[24]. Taken together, we believe that the biological properties of ucenprubart support further testing in clinical studies.

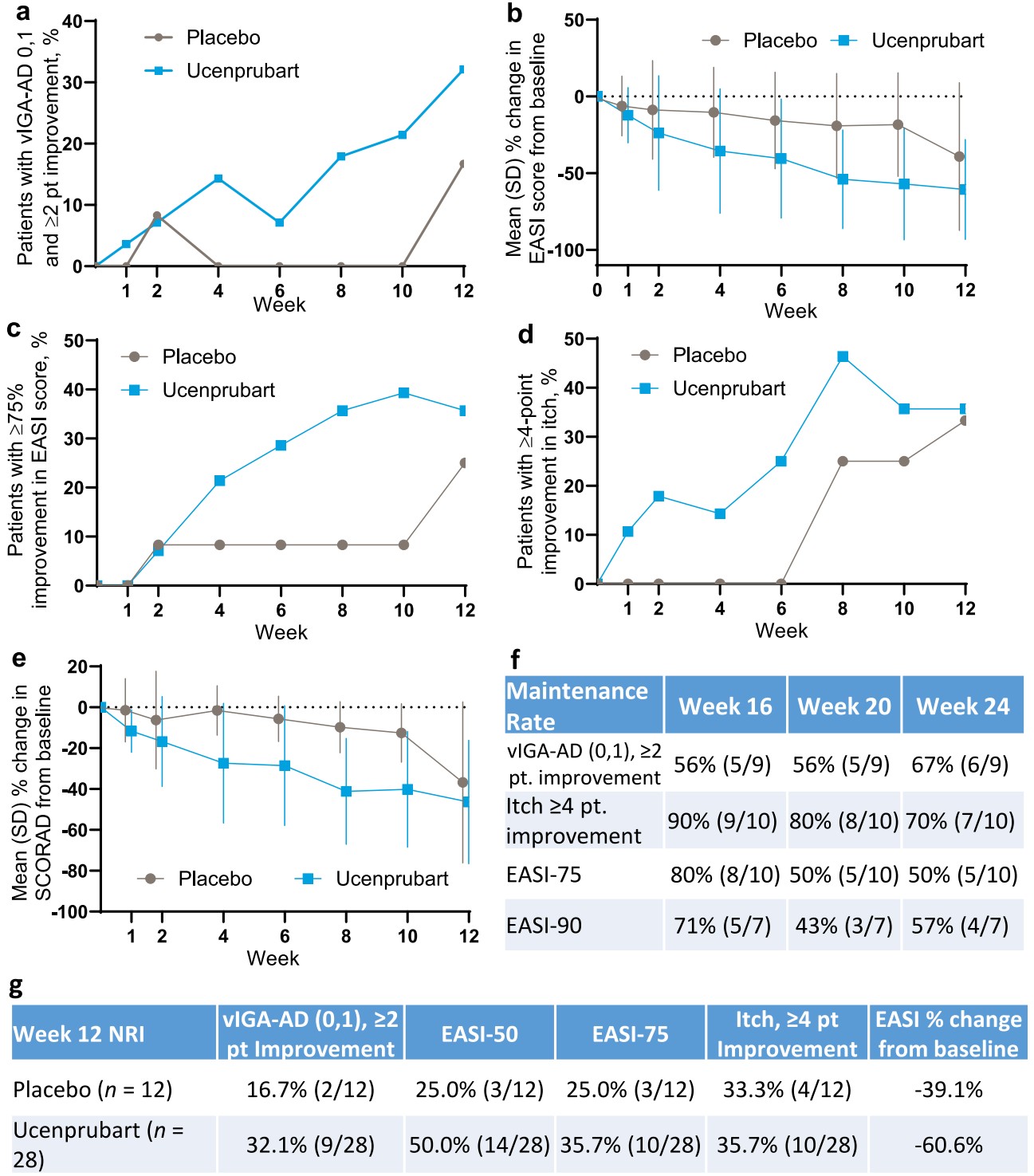

**Fig. 5 | Phase 1 clinical efficacy results in patients with atopic dermatitis.**
**a** Proportion of patients who met the vIGA-AD endpoint (0,1 score and ≥2-point improvement from baseline) by study week, including the 12-week follow-up period. **b** Percentage change in EASI score from baseline to week 12 (overall mean percentage change in EASI score between ucenprubart and placebo, MMRM $p = 0.0157$). **c** Proportion of patients with a ≥75% improvement in EASI score (EASI-75) from baseline to week 12. **d** Proportion of patients with a ≥4-point improvement in itch (SCORAD pruritis component) from baseline to week 12. **e** Percentage change in the SCORAD score from baseline to week 12 (overall mean percentage change in SCORAD between ucenprubart and placebo, MMRM $p = 0.004$).

**f** Maintenance response rates during follow-up for ucenprubart recipients who met vIGA-AD (0,1) with a ≥2 point improvement, ≥4 point improvement in itch, or EASI ≥ 75% improvement (EASI-75 and ≥90% improvement in EASI score [EASI-90] maintenance) at week 12. **g** Response rates at week 12 for placebo and ucenprubart. Missing data were handled using NRI. EASI eczema area and severity index, MMRM mixed model for repeated measures, NRI nonresponder imputation, pt point, SCORAD SCORing atopic dermatitis, SD standard deviation, vIGA-AD validated investigator global assessment for atopic dermatitis. Source data is provided as a Source Data file.

**Table 1 | Safety summary in Part C**

| Category | Placebo IV (N = 12) | Ucenprubart 500 mg IV (N = 28) |
|---|---|---|
| All TEAEs, no. of events [no. of pts] (% of pts) | 7 [5] (41.7) | 25 [11] (39.3) |
| TEAEs by intensity | | |
| Mild | 6 | 13 |
| Moderate | 1 | 12 |
| Severe | 0 | 0 |
| TEAEs in ≥2 pts, no. of events [no. of pts] (% of pts) | | |
| Contusion | 0 | 2 [2] (7.1) |
| Dermatitis contact | 0 | 3 [2] (7.1) |
| Urinary tract infection | 0 | 2 [2] (7.1) |
| AEs leading to discontinuation | 0 | 0 |
| Serious AEs | 0 | 0 |

Data are shown as no. of events unless otherwise indicated.
*AE* adverse event, *IV* intravenously, *no.* number, *pts* patients/participants, *TEAE* treatment-emergent adverse event.

In phase 1 clinical testing, we found an acceptable safety profile for ucenprubart in healthy participants after single or repeat dosing. Those findings were maintained in patients with AD, with no serious AEs, discontinuations due to AEs, or clinically significant vital sign or clinical laboratory findings through 12 weeks of dosing. Statistically significant differences in overall mean percentage change from baseline versus placebo were seen in EASI and SCORAD scores through 12 weeks. Among patients who met various response criteria at week 12, at least half maintained that response at 12 weeks of drug-free follow-up, consistent with the possibility of durable efficacy in week 12 responders. Additionally, our ex vivo assessment of lesional skin biopsies following CD200R agonism found reductions in CD3 and GATA3 staining in the dermis from EASI-75 responders, consistent with a reduction in activated Th2 cells. We also observed improvements in the structure and organization of the stratum spinosum and basal layers based on histological findings. Collectively, these findings show a clear association between clinically assessed skin lesion improvement based on changes in EASI and SCORAD scores, and changes in Th2 cell differentiation markers in skin tissue based on histology following agonism of the CD200R checkpoint receptor.

We next sought to explore downstream pathways that were affected by CD200R activation and to assess potential differences relative to an existing treatment in AD using a humanized mouse model of contact dermatitis. Dupilumab, an anti-IL-4Rα antibody that blocks IL-4 and IL-13 signaling, was approved for treatment of moderate to severe eczema (AD) in the United States in 2017[25]. Using CITE-seq to identify and then compare differentially regulated genes following dupilumab versus ucenprubart treatment (IL-4 receptor inhibition vs CD200R agonism), we observed a notably different profile between the 2 treatments. Most gene expression changes with ucenprubart were found in myeloid cells, particularly macrophages. In contrast, dupilumab treatment induced gene changes in a broader set of cells, including various T-cell types in addition to macrophages. Importantly, we were able to confirm previously reported changes in immune pathways following dupilumab treatment, including changes in Th2- and Th17-associated signaling pathways in CD4+ T cells[26,27]. In contrast, using pathway analysis for ucenprubart treatment we identified several immune-related pathways that were modulated in macrophages. These pathways seem to indirectly affect Th2 cell retention in the skin of responders, suggested by a numerically greater decrease in GATA3 signaling using immunohistochemistry. Among the top pathways specifically enriched with ucenprubart, several were linked to macrophage function including NF-kappa B signaling and JAK-STAT signaling. The NF-kappa B axis provides a mechanism for macrophages

to balance pro- and anti-inflammatory responses[28]. Additionally, non-canonical NF-kappa B signaling is important for the function of highly suppressive regulatory T (T_reg) cells, and leads to the activation and proliferation of these cells[29]. Furthermore, both NF-kappa B and JAK-STAT signaling in macrophages have been intrinsically linked to TNF activation in AD[30,31]. Notably, eczema is a common adverse effect of anti-TNF-α therapy that has been reported in up to 20% of patients receiving long-term treatment for psoriasis, arthritis, or inflammatory bowel disease[32–34]. While we did not observe modulation of T_reg cells with ucenprubart in the humanized mouse model, the modulation of myeloid-derived suppressor cells and T_reg cells is associated with immunosuppressive activity[35]. Effects on such cell types could lead to modulation of the inflammatory state of the tissue environment and hence have an indirect effect on the retention of Th2 cells in the skin of ucenprubart responders. Therefore, we speculate that the observed efficacy of ucenprubart was mostly macrophage driven, which could also explain the lack of modulation of the classical Th2 cytokines associated with AD in ucenprubart responders. Further study of these pathways in skin samples from patients with AD treated with ucenprubart is needed.

There are several limitations to the interpretation of the present analyses. For the CITE-seq analysis of cell types with differential expression and pathway analysis, data were generated based on human cells in a humanized mouse model of skin inflammation, with cells from a single donor. In addition, the engraftment and development of human cells in the humanized mouse model may not represent all cell types found in human skin; hence, rare cell types may have been missed. Given the small sample size that increases the risk of false discoveries when reporting individual gene results[36], we centered our analysis on gene set enrichment and correlation approaches that obtain more robust results than single gene tests[37]. Further studies with larger sample sizes are needed, including further biopsy and immunohistochemistry assessment of lesional samples from patients treated with ucenprubart, to elucidate possible contributions to clinical improvements in patients with AD. In addition, AD is heterogenous in both pathophysiology and presentation and involves complex interactions between epidermal barrier dysfunction, immune activation, the skin, and potentially gut microbiomes[14,38–41]. The complexity and incomplete understanding of these interactions limit our ability to interpret some findings on clinical biomarkers, particularly for a therapy with a different mechanism of action. Similarly, various AD endotypes have been identified[40,42,43] that could show variable response to some treatments. It will be of interest in future studies to assess additional cellular and other markers in patients as predictors of response to ucenprubart therapy. Finally, rapid AD recurrence is common following withdrawal of anti-IL4-Rα therapy[44,45]. Given the substantially different mechanism of action and the relative durability of effects seen in week 12 responders through the 12-week drug-free follow-up period, further study is needed to assess the potential for rebalancing immune homeostasis following immune checkpoint therapy with ucenprubart.

In conclusion, the preclinical and clinical data for the CD200R agonist ucenprubart demonstrate that targeting the immune checkpoint inhibitory receptor CD200R provides opportunities for durable treatment of autoimmune or inflammatory diseases such as AD. The present findings support continued testing of ucenprubart in clinical trials among patients with AD (NCT05911841).

## Methods

### Antibody generation

LY3454738 (ucenprubart) was derived from a rabbit mAb discovered from an alternating soluble ECD human CD200R-IgG1 Fc/cynomolgus CD200R-IgG1 Fc protein immunization in rabbits. CD200R (receptor) proteins for immunization and screening were generated through gene synthesis and cloning into mammalian expression vectors as

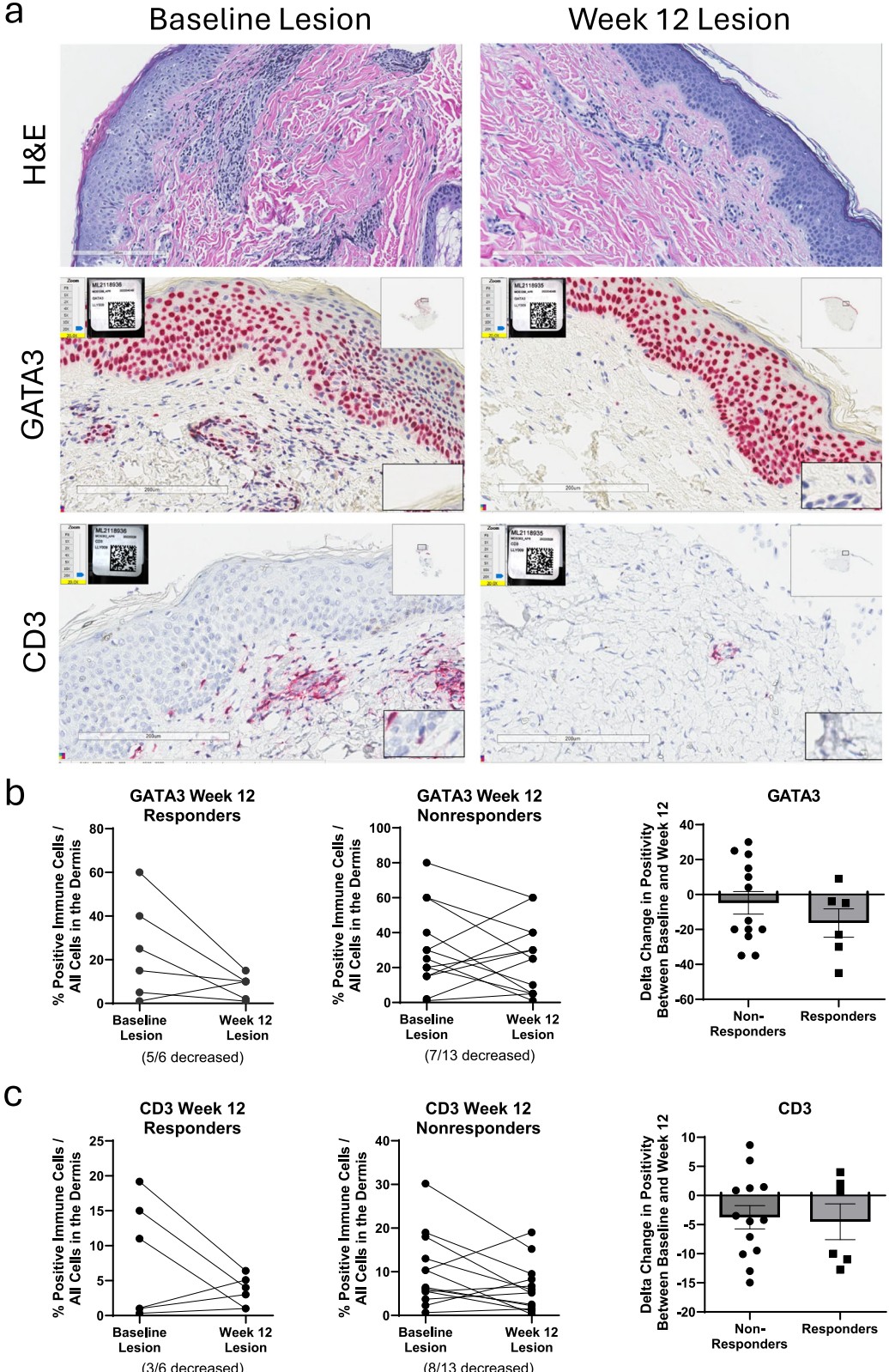

**Fig. 6 | CD3 and GATA3 are reduced from baseline in week 12 responders.**
**a** Baseline and week 12 tissue biopsies were treated with H&E and immunohisto-chemical staining for the expression of GATA3 and CD3. Scale bar = 200 μm. The percentage of positive immune cells in the dermis and change from baseline are shown by EASI-75 week 12 responder status ($n = 6$ for responders and $n = 13$ for non-responders) for **b** GATA3[+] and **c** CD3[+]. EASI-75, ≥75% improvement in the eczema area and severity index score; GATA3, GATA binding protein 3 (a marker for Th2 cells); H&E hematoxylin and eosin. Source data is provided as a Source Data file.

N-terminal factor Xa IgG1 Fc fusions or as octa-His-tagged proteins. All sequences were gathered from the National Center for Biotechnology Information (NCBI), except for cynomolgus CD200R, which was generated in house, and cyCD200RLa (cynomolgus-activating form) sequence, which was obtained via a patent publication (WO08079352). Allele 1 and allele 2 are the 2 predominant CD200R alleles in the human population (UniProtKB; Q8TD46: MO2R1_HUMAN). At the end of immunization, rabbits were bled, and the plasma was tested for binding to human CD200R and cynomolgus CD200R. Flow cytometry was used for single-cell sorting of IgG-positive B cells from 4 different rabbits into separate 96-well plates. Each cell/well was lysed, and single-cell polymerase chain reaction was used to produce variable domain fragments that were cloned into DNA plasmids encoding rabbit heavy chain or light chain constant regions. Plasmids for each rabbit antibody heavy chain and light chain were mixed and transfected into HEK293 cells for transient expression. After 6 days, 96-well plates harboring the mAb library were collected for screening against the following soluble fusion proteins: human CD200Ra1-IgG1 Fc (allele 1), human CD200Ra2-IgG1 Fc (allele 2), cynomolgus CD200R-IgG1 Fc, CD200RLa-IgG1 Fc, mouse CD200R-IgG1 Fc, a CD200/CD200R fusion protein, and an unrelated IgG1 Fc control protein. Selected antibodies were cross-reactive with human and cynomolgus CD200R but did not bind CD200RLa and unrelated IgG1 Fc. Four rabbit antibodies were chosen for humanization including rabbit parental LY3454738. The top 4 framework pairings with the best cellular CD200R binding were scaled-up, purified, and assessed for their binding affinity using surface plasmon resonance. The humanized parental LY3454738 variant chosen based on these analyses had a 1-46 VH/A26 VL germline framework pairing and a final affinity slightly stronger than what was measured for the parental rabbit antibody. LY3454738 was expressed in Chinese hamster ovary (CHO, ATCC catalog (Cat) # CCL-61) cells. Antibodies used in the preclinical studies were purified by protein A affinity chromatography and size exclusion chromatography. The CD200 protein ligand was also generated to evaluate each antibody's ability to block the CD200 ligand from binding to CD200R.

## Surface plasmon resonance

The Biacore T100 instrument (Cytiva), Biacore reagents, and Scrubber2 Biacore Evaluation Software (BioLogic Software 2008) were used for the surface plasmon resonance analysis of LY3454738 binding. A Series S Sensor Chip CM4 (Cytiva, Cat #BR100534) was prepared using the 1-ethyl-3-(3-dimethylaminopropyl) carbodiimide/N-hydroxysuccinimide (EDC/NHS) amine coupling method of the manufacturer (Cytiva, Cat #BR100050). Briefly, the surfaces of all 4 flow cells (FCs) were activated by injecting a 1:1 mixture of EDC and NHS for 7 min at 10 μL/min. Protein A from Staphylococcus aureus (Calbiochem, Cat #539202) was diluted to 100 μg/mL in 10 mM acetate buffer (pH 4.5) and immobilized for approximately 400 resonance units (RU) onto all 4 FCs by a 7-min injection at a flow rate of 10 μL/min. Unreacted sites were blocked with a 7-min injection of ethanolamine at 10 μL/min. Glycine injections of $2 \times 10$ μL (pH 1.5) were used to remove any non-covalently associated protein. The running buffer was 1× HBS-EP+ (Cytiva, Cat #BR100669).

Extracellular and monomeric human, cynomolgus, and cynomolgus-activating CD200R proteins were purified in house using immobilized metal affinity chromatography and size exclusion chromatography. For human and cynomolgus CD200R binding, antibodies were diluted to 2.5 μg/mL in running buffer, and approximately 150 RU of LY3454738 was captured in FC2 through FC4 ($RU_{captured}$). FC1 was the reference FC; therefore, no antibody was captured in FC1. Human and cynomolgus CD200R were diluted to 500 nM in running buffer and then 2-fold serially diluted in running buffer to 3.9 nM. Duplicate injections of each concentration were injected over all FCs at 50 μL/min for 250 s followed by a 1200-s dissociation phase. Regeneration

was performed by injecting 15 μL of 10 mM glycine (pH 1.5) at 30 μL/min twice over all FCs. Reference-subtracted data were collected as FC2-FC1, FC3-FC1, and FC4-FC1. The measurements were obtained at 37 °C. The affinity ($K_D$) was calculated using a 1:1 Langmuir-binding model in BIAevaluation (BiacoreAD)

For cyCD200RLa binding, antibodies were diluted to 2.5 μg/mL in running buffer, and approximately 150 RU of LY3454738 was captured in FC2 through FC4 ($RU_{captured}$). FC1 was again the reference FC. CyCD200RLa was diluted to 8.1 μM in running buffer and then 2-fold serially diluted in running buffer to 63.2 nM. Duplicate injections of each concentration were injected over all FCs at 50 μL/min for 250 s followed by a 1200-s dissociation phase. Regeneration was performed by injecting 15 μL of 10 mM glycine (pH 1.5) at 30 μL/min twice over all FCs. Reference-subtracted data were collected as FC2-FC1, FC3-FC1, and FC4-FC1. The measurements were obtained at 37 °C. The affinity ($K_D$) was calculated using the steady-state equilibrium analysis with the Scrubber2 Biacore Evaluation Software.

The FcγR ECDs FcγRI (CD64), FcγRIIa-131 R/H (CD32A), FcγRIIIa-158 V/F (CD16A), and FcγRIIb (CD32B) were produced from stable CHO cell expression. All FcγR ECDs were purified using IgG Sepharose and size exclusion chromatography. For FcγRI binding, antibodies were diluted to 2.5 μg/mL in running buffer, and approximately 150 RU of each variant was captured in FC2 through FC4 ($RU_{captured}$). FC1 was the reference FC; therefore, no antibody was captured in FC1. The FcγRI ECD was diluted to 200 nM in running buffer and then 2-fold serially diluted in running buffer to 0.78 nM. All FCs were injected with at least duplicate injections of each concentration at 40 μL/min for 120 s followed by a 1200-s dissociation phase. Regeneration was performed by injecting 15 μL of 10 mM glycine (pH 1.5) at 30 μL/min over all FCs. Reference-subtracted data were collected as FC2-FC1, FC3-FC1, and FC4-FC1. The measurements were obtained at 25 °C. The affinity ($K_D$) was calculated using either the steady-state equilibrium analysis with the Scrubber2 Biacore Evaluation Software or a 1:1 Langmuir-binding model in BIAevaluation. For FcγRIIa, FcγRIIb, and FcγRIIIa binding, antibodies were diluted to 5 μg/mL in running buffer, and approximately 500 RU of each variant was captured in FC2 through FC4 ($RU_{captured}$). FC1 was again the reference FC. Fcγ receptor ECDs were diluted to 10 μM in running buffer and then 2-fold serially diluted in running buffer to 39 nM. Duplicate injections of each concentration were injected over all FCs at 40 μL/min for 60 s followed by a 120-s dissociation phase. Regeneration was performed by injecting 15 μL of 10 mM glycine (pH 1.5) at 30 μL/min over all FCs. Reference-subtracted data were collected as FC2-FC1, FC3-FC1, and FC4-FC1. The measurements were obtained at 25 °C. The affinity ($K_D$) was calculated using the steady-state equilibrium analysis with the Scrubber2 Biacore Evaluation Software.

## In vitro determination of cell binding by ucenprubart

CHO cells were stably transfected with human CD200R, cynomolgus CD200R, or cynomolgus CD200RLa, and non-transfected CHO cells were used as a negative control. Cells were prepared with $1 \times 10^5$ cells/well/cell line (100 μL) incubated for 30 min at room temperature with 10-point ucenprubart titration in duplicate prepared in fluorescence-activated cell sorting (FACS) buffer. For confirmation of cyCD200RLa expression, cells were stained with cy-La antibody specific for CD200RLa (generated by Eli Lilly). The cells were washed twice with FACS buffer before adding 100 μL of cocktail with phycoerythrin (PE)-conjugated antihuman Fc antibody (Jackson Immunoresearch) at a 1:1500 dilution and eFluor 780 live/dead stain at 1:2000 for 15 min at 4 °C in the dark. The cells were washed twice and then resuspended in 125 μL of FACS buffer to run on the Cytek Aurora flow cytometer.

## Co-binding flow cytometry experiment

HEL 92.1.7 cells (American Type Culture Collection [ATCC], catalog No. TIB-180) were incubated for 2 h with 66 nM ucenprubart or CD200 Fc or isotype control at room temperature. Cells were then Fc blocked for

20 min and incubated with different concentrations of ucenprubart AF647 for 1 h at room temperature in the dark. The cells were run on a BD LSR II Flow Cytometer (BD Biosciences), and 10,000 cells were collected off the second single-cell gate for analysis.

## A20:DO11.10 assay

DO11.10 is a T-cell hybridoma that responds to antigenic stimulation with ovalbumin peptide 323–339 by secreting IL-2. DO11.10 cells (Lonza) were transfected with human CD200R to use in a reporter assay. Human DO11.10-CD200R cells were maintained in culture at concentrations of $0.2 \times 10^5$ to $2 \times 10^6$ cells/mL in RPMI 1640 medium containing 10% heat-inactivated fetal bovine serum (FBS), 1× MEM nonessential amino acids, 1× GlutaMAX, 1× sodium pyruvate, 50 mM HEPES, 1 μg/mL puromycin, and 1× MEM vitamin solution for less than 21 days. A20 (ATCC, catalog No. TIB-208), a mouse B-cell lymphoma line, was used to stimulate DO11.10 cells in this assay. Human CD200 was transfected into the A20 cells. Human A20-CD200 cells were maintained in culture medium at concentrations from $0.2 \times 10^5$ to $2 \times 10^6$ cells/mL in RPMI 1640 containing 10% heat-inactivated FBS, 1× β-mercaptoethanol, 1× GlutaMAX, and 0.5 mg/mL geneticin (G418) for less than 21 days. Both cell lines were collected individually into 50-mL Falcon tubes and centrifuged at $600 \times g$ for 5 min at room temperature. Supernatant was aspirated and each cell pellet was resuspended at $1 \times 10^6$ cells/mL in RPMI 1640 medium supplemented with 10% heat-inactivated FBS, 1× β-mercaptoethanol, and 1× Gluta-MAX. Human DO11.10-CD200R cells of $5 \times 10^5$ cells/well, followed by $5 \times 10^5$ cells/well of human A20-CD200 cells, were seeded into tissue culture-treated U-bottom 96-well plates using a Thermo Scientific Multidrop Combi Reagent Dispenser. Plates were incubated at room temperature during treatment preparation. Ucenprubart was prepared at 24 μM in Dulbecco's phosphate-buffered saline (DPBS). DPBS was used as the untreated control. A 10-point 3-fold dilution series was prepared for ucenprubart in DPBS, after which assay medium was added to the dilution series and untreated control (3× dilution). Finally, 50 μL of the dilution series and untreated control were added in duplicate to the 100 μL of seeded cells. Prepared 1-mg/mL aliquots of ovalbumin peptide (323–339) (chicken, Japanese quail; Sigma), resuspended in distilled water, were thawed on ice. Ovalbumin peptide was prepared at 1 μg/mL in assay medium. Then, using the Thermo Scientific Multidrop Combi Reagent Dispenser, 50 μL of ovalbumin peptide was added to every well of the assay plate containing cells and antibody treatment. Plates were sealed with AeraSeal film (Excel Scientific) and incubated for 18 h at 37 °C in a humidified environment with 5% carbon dioxide ($CO_2$), after which 2 μL of each supernatant was collected. Mouse IL-2 concentrations were determined using the AlphaLISA Mouse IL-2 Assay Kit (PerkinElmer) with readouts measured on a BioTek Synergy Neo2 (Agilent BioTek) multimode plate reader. IL-2 levels were converted to percent inhibition using the untreated control as the baseline, and a 4-parameter logistic model was used to fit the data (R statistical software).

## U937 IL-8 assay

U937 cells, a monocytic cancer cell line (ATCC, catalog No. CRL-1593.2), were transfected with complementary DNA for human CD200R. For FcγR-stimulated cytokine release, high-binding plates were coated with 10 μg/mL of human IgG1 isotype antibody in 100 μL of DPBS or DPBS only. Cells were preincubated with DPBS, isotype control, or ucenprubart at different concentrations before being plated on human recombinant IgG1 (produced in house)-coated 96-well plates and incubated for 24 h in 10 ng/mL IFN-γ medium. After incubation, the supernatant was harvested, and samples were tested at a 1:10 dilution for IL-8 concentrations using the Meso Scale Discovery IL-8 kit (Meso Scale Discovery) according to the manufacturer's instructions.

## Staphylococcal enterotoxin B T-cell proliferation assay

Human peripheral blood mononuclear cells were isolated from healthy donors and stained for 30 min with 1 μM carboxyfluorescein succinimidyl ester (Biolegend, Cat #79898) at room temperature. Stained cells were distributed into 96-well culture plates ($1 \times 10^5$ cells/well) with preloaded treatment antibodies at 30 μg/mL in RPMI 1640 medium with 10% heat-inactivated FBS and 1% penicillin/streptomycin. Cells were then stimulated with 8 ng/mL staphylococcal enterotoxin B for 96 h at 37 °C in a humidified environment with 5% $CO_2$. After incubation, CFSE labeled cell plates were washed with FACS buffer (Miltenyi Biotec; Cat #130-091-221), incubated with human Fc blocking reagent (Miltenyi Biotec, Cat #130-059-901) for 10 min followed by staining with labeled antibodies against CD4 AF700 (Biolegend, Cat #317426) and PD-1 PE (eBioscience, Cat #17-2799-42) for 30 min on ice. After washing in FACS buffer, live/dead cell staining was performed using SytoxBlue (Invitrogen, Cat #S34857) per manufacturer's protocol. Cells were processed on Fortessa X20 flow cytometry instrument in the presence of CountBright Beads (Invitrogen, Cat #C36950) to quantitate the cell number. Flow cytometry data were analyzed using FlowJo software. Absolute dividing CD4 positive cell count was calculated using CFSE staining and CountBright beads (see FACS gating strategy in Supplementary Fig. 5). Data were collected from three individual donors.

## Whole blood cytokine release assay

Whole blood samples were obtained from healthy donors after approved consent in accordance with the ethical practice of the research biological donation program committee of Eli Lilly and Company.

The whole blood assay was performed to assess cytokine release risk in response to mAbs[21]. Briefly, heparinized whole blood was obtained from healthy donors and used within 2 h after donation. Blood (225 μL) was added into 96-well plates. LY3454738 was generated as three isotypes with different FcγR-binding properties and evaluated at 100 μg/mL: IgG4 S228P (IgG4P; ucenprubart)[46], IgG4P F234A/L235A (IgG4PAA)[47], and IgG1 (Supplementary Table 2). The negative control, evaluated at 100 μg/mL, was an IgG1 antibody that does not induce cytokine release in the clinic. The positive control, evaluated at 10 μg/mL, was a homolog of alemtuzumab (anti-CD52 IgG1, Eli Lilly generated) known to cause cytokine release syndrome. mAbs were diluted in DPBS and added to the blood at 25 μL. Additionally, cytokine levels in whole blood cells treated with DPBS were used as the donor baseline. After 24-h incubation at 37 °C in a humidified environment with 5% $CO_2$, plates were centrifuged at $400 \times g$ and blood plasma supernatants were collected. Cytokine levels in plasma supernatants were detected using the V-PLEX Proinflammatory Panel (human) Kit assay (Meso Scale Discovery, catalog no. K15049D-2) following the manufacturer's instructions.

## Contact dermatitis in humanized mice

Ethical use of animals: The use of mice followed the ethical guidelines stated in the National Institutes of Health Guide for the Care and Use of Laboratory Animals. Animal study protocols were approved by the Animal Policy and Welfare Committee of Eli Lilly and Company.

Female huNOG-EXL mice (NOD.Cg-*Prkdc*[scid] *Il2rg*[tm1Sug] Tg(SV40/HTLV-IL3,CSF2)10-7Jic/JicTac, model 13395-F, Taconic) were purchased at 20 weeks of age and allowed to acclimate for >1 week. Mice were housed at 4 per cage at 22 °C under a 12-h light:dark cycle and allowed food and water ad libitum. Mice were housed in compliance with local IACUC guidelines in microisolator cages in a barrier facility. Cages, bedding, food and water were sterilized. Mice were randomized by bodyweight into groups (isotype, $n = 9$; dupilumab, $n = 8$; ucenprubart, $n = 6$) and housed four mice per cage with mice from different treatment groups randomized to the cage. On day 0, mice were anesthetized with 5% isoflurane, their abdomens shaved, and 100 μL of

3% oxazolone in ethanol was applied to the shaved area (sensitization). To elicit inflammation, mice were anesthetized on day 5 with 5% isoflurane, their baseline ear thickness was measured with calipers, and 10 μL of 2% oxazolone in ethanol was applied on each side of both ears (challenge). The challenge procedure was repeated on days 11 and 19. The hypersensitivity reaction was assessed by measuring the difference between ear thickness before the first challenge and 24 h after each challenge. Ears were harvested 1 day after the third challenge for CITE-seq analysis. A single dose of ucenprubart was administered at 10 mg/kg SC 24 h before the first challenge; IgG4P isotype or dupilumab (produced recombinantly in house) was administered at 10 mg/kg SC before sensitization and each challenge for comparison. Effects on the inflammatory response were calculated as change in ear thickness from baseline and considered significant if the $p$-value was <0.05 by 1-way analysis of variance with the Tukey post hoc test (GraphPad Prism).

### CITE-seq experimental protocol

After euthanasia by asphyxiation with carbon dioxide followed by cervical dislocation, both ears of 2–3 mice within the same treatment group were pooled into 2 biological replicates for digestion using the Multi Tissue Dissociation Kit 1 (Cat #130-101-540, Miltenyi Biotec) according to manufacturer's recommendations in the Dissociation of Mouse Ear protocol. The resulting single-cell suspensions were incubated in Cell Staining Buffer (Cat #420201, BioLegend) containing Human TruStain FcX (Cat #422305, BioLegend) on ice for 15 min and protected from light. Afterwards, Live/Dead Fixable Near-IR stain (Invitrogen), Alexa Fluor 700-conjugated antihuman CD45 (clone 2D1; Cat #47-0459-41, eBioscience), antihuman CD200R (clone OX-108; Cat #329307, BioLegend) that was biotinylated using the EZ-Link Sulfo-NHS-LC-Biotin kit (Thermo Scientific, Cat #21343), and TotalSeq-B-conjugated antihuman antibodies (CD3, CD4, CD8a, CD20, CD33, CD123, and CD200R-Biotin; BioLegend) were added for 30 min on ice and protected from light. (TotalSeq-B antibody clone number, BioLegend catalog #, and final concentration in μg/mL, respectively. were as follows: CD3: UCHT1, 300477, and 0.25; CD4: RPA-T4, 300565, and 0.0625; CD8a: RPA-T4, 300565, and 0.0625; CD20: 2H7, 302361, and 0.25; CD33: P67.6, 366635, and 0.5; CD123: 6H6, 306047, and 1.00; and CD200R-Biotin: OX-108, 329302, and 5.)

Cells were washed twice in Cell Staining Buffer before incubation with TotalSeq-B-PE-conjugated Streptavidin (BioLegend) diluted in Cell Staining Buffer for 20 min on ice and washed twice in Cell Staining Buffer. Cells were diluted in Cell Staining Buffer to $2 \times 10^7$ cells/mL (500 μL minimum volume) and human CD45$^+$ cells collected in RPMI 1640 medium supplemented with 50% FBS using the BD FACSAria III Cell Sorter (BD Biosciences; see Supplementary Fig. 5). If yields were higher than 20,000 cells after cell sorting, cells were counted and diluted to $1 \times 10^6$ cells/mL in DPBS and 10,000 cells from each sample were loaded onto separate lanes of a Chromium Controller (10× Genomics). For samples with yields lower than 20,000 cells, the entire sample was loaded onto an individual lane of a Chromium Controller. The resulting gel beads-in-emulsion were processed according to the standard single cell 3' v3.1 assay protocol from 10× Genomics. After library preparation and i7 indexing of individual samples, the final pooled library comprised 90% messenger RNA and 10% antibody-derived tags. Libraries were sequenced on the Illumina NovaSeq 6000 sequencing system (Azenta) to generate 150 base paired-end reads. Quantification was done with the Cell Ranger v6.1.2 (10× Genomics) xenograft pipeline against mouse (mm10) and human (HGRC38) genomes.

### Single-cell clustering, cell type annotation, and statistical analysis

For CITE-seq analyses, the Seurat R package (v5.1.0) was used to perform clustering of cells derived from dupilumab- and ucenprubart-treated mouse ears. Samples were merged into a single expression matrix using the merge function. The data were submitted to the NCBI Gene Expression Omnibus and can be retrieved under accession number GSE220685. Only cells derived from the humanized mouse model harboring transcripts that uniquely mapped to the human genome (HGRC37 built) were retained for downstream analysis and cell clustering. Low-quality human cells with <200 transcripts, <100 genes, or >25% of mitochondrial gene expression were filtered out. The NormalizeData function in Seurat was used to normalize gene expression levels for each cell with default parameters. In addition, the ScaleData function was used to scale and center the RNA and protein surface marker counts in the data set. To account for potential batch effects, principal component analysis was performed on the most variable genes after regressing out effects of cell cycle genes. The first 30 principal components were then used to remove batch effects from samples processed in different libraries. UMAP dimensional reduction was performed using the RunUMAP function with default parameters. To generate cell clusters, the FindNeighbors and FindClusters functions were used with a resolution set to 0.5. Seurat cell clusters derived from RNA sequencing data were annotated using the SingleR package (v3.15)[48]. In addition, cell types were manually curated by overlapping the cluster markers identified by the FindMarkers function with canonical cell type signature genes from the protein atlas (https://www.proteinatlas.org). Broader cell type categories were further validated using a combination of CITE-seq surface markers (Supplementary Fig. 1). Differential expression analysis between treatments and isotype controls were performed using the nonparametric Wilcoxon rank-sum test implemented in the FindMarkers function. Genes with a Bonferroni-corrected $p < 0.05$ were considered differentially expressed across treatments (Supplementary Table 3). Significant enrichment of KEGG and Reactome pathways among differentially expressed genes was determined using a hypergeometric distribution test implemented in the R clusterprofiler package[49]. Functional pathway enrichment was performed using the GSEA method developed by Subramanian et al.[37] implemented in the FGSEA package (v3.15). Briefly, ranked gene lists were generated for each treatment by ranking genes by fold change compared with isotype controls. Enrichment analysis was performed for curated gene sets covering hallmark pathways from the Molecular Signatures Database (v7.5.1; https://www.gsea-msigdb.org/gsea/msigdb/), and significance was determined after false discovery rate multiple testing correction.

### Phase 1 clinical study

The protocol for Study J1B-MC-FRCC (NCT03750643) is provided as Supplementary Appendix 2. Briefly, this was a phase 1, multicenter, randomized, placebo-controlled, triple-blind study designed in three parts with staggered initiation. The study assessed single-ascending dose and repeat-dose arms in healthy participants and a repeat-dose arm in participants with AD to explore the safety, tolerability, pharmacokinetics, target engagement, and, in participants with AD, clinical pharmacodynamics and efficacy of ucenprubart.

Part A had a single-ascending dose design and included healthy participants who received ucenprubart at 1, 5, 25, 100, 300, and 1000 mg or placebo IV and ucenprubart at 100 mg SC. Part B had a repeat-dose design and included healthy participants who received ucenprubart at 2 doses of 200 mg or placebo IV at a 2-week interval. Part C had a repeat-dose design and included patients with AD who received ucenprubart at 500 mg or placebo IV every 2 weeks for 12 weeks. Participants in all study parts were followed up for 12 weeks after the last dose of blinded study treatment. Parts A and B were conducted in a confined clinical research unit, while Part C was conducted in an ambulatory setting. Because the study overlapped with the COVID-19 epidemic, 10 additional patients were added to Part C due to concerns about the potential for an elevated dropout rate and/or slow enrollment at existing sites.

Individual participants were enrolled into a single part of the study only. In Part A, participants were enrolled to only 1 cohort. In Part A, sentinel ucenprubart dosing was used in Cohorts 1 and 2 for 2 participants (placebo [$n = 1$]; ucenprubart [$n = 1$]). Each IV cohort in all parts of the study included treatment-matched placebo via the same route of administration as that used for ucenprubart. Participants were randomized 3:1 to study drug versus placebo in Parts A and B and 2:1 to study drug versus placebo in Part C. In a single SC dosing subcohort in Part A, 6 participants received 100 mg of ucenprubart. In each part, the final dose was followed by a 12-week follow-up period.

**Study populations.** The study enrolled male and female participants aged ≥18 (≥20 for Japanese participants) to ≤65 years at screening with a body mass index of ≥18 to ≤32 kg/m² (Parts A and B) or ≥18 to ≤45 kg/m² (Part C) and body weight ≥50 kg. Participants in Part C were diagnosed with AD, as defined by the American Academy of Dermatology[50], at least 12 months before screening and had moderate to severe AD at screening and randomization, including an EASI score ≥12, a vIGA-AD score ≥3, and ≥7% of body surface area involvement. In addition, participants must have applied emollients daily for at least 14 days before randomization and agreed to use emollients daily throughout the treatment period.

Individuals were excluded from study enrollment if they were immunocompromised, had skin conditions that could interfere with assessments, or had infections at screening. Additional exclusion criteria that applied to individuals in Parts A and C are shown in the protocol. This study was conducted in accordance with the ethical principles of the Declaration of Helsinki and Good Clinical Practice guidelines and approved by the institutional review board or ethics committee for each center. All participants provided written informed consent before the first study procedure. The study protocol was approved by the Institutional Review Board at Advarra, with an redacted version along with the statistical analysis plan found at the end of the Supplementary Information PDF file.

**H&E staining and immunohistochemistry.** Four-mm punch biopsies were obtained from lesional skin at baseline and week 12. Samples were fixed in formalin and embedded in paraffin wax. Tissues were then sectioned at 4 microns and attached to glass slides for immunohistochemistry or H&E staining using standard protocols. For immunohistochemistry, specifically, tissue sections were pretreated with BOND Epitope Retrieval Solution 2 (Leica, catalog No. AR9640) before the addition of mAbs against CD3 (clone LN10; Leica) and GATA3 (clone L50-823; Abcam) using the Leica BOND RX autostainer with CDX-capable protocols with red chromogen. Representative images (20×) were taken with standard pathology interpretation performed. For CD3, image analysis was performed on the dermis, excluding the epidermis and adnexal structures of the skin. GATA3 positivity was determined on the same region of interest by two pathologists scoring the percentage of positive immune cells among all cells in the dermis.

**Sample size considerations.** The sample sizes in Part A (54 participants) and Part B (8 participants) were not based on statistical calculations, as these cohorts were designed primarily to seek information on safety, tolerability, pharmacokinetics, and target engagement. The sample sizes were not adjusted to account for dropouts. However, the protocol allowed for replacement if it was deemed necessary to obtain sufficient data for interpretation.

For Part C, assuming 50% of participants receiving LY3454738 and 8% of participants receiving placebo would achieve a vIGA-AD 0/1, a sample size of 30 completers would provide approximately 70% power using a 2-sided Fisher's exact test at the 0.10 significance level. Participants who were randomized but not administered treatment could have been replaced to ensure that approximately enough participants completed the study. The study was conducted from first participant

first visit 29 November 2018 to last participant last visit on 07 September 2021.

Note that the number of participants listed at clinicaltrials.gov as of 11 December 2024 is a preliminary number that will be updated after the compound has received marketing approval or has been terminated. At the time of this writing, the number listed at clinicaltrials.gov was 128[51]; however, the correct number per the trial data is 102.

**Per protocol efficacy endpoints.** The primary efficacy endpoint in Part C was the binary outcome of response defined as a vIGA-AD score of 0 or 1 (clear or almost clear skin) with a ≥2-point improvement from baseline at week 12.

Secondary efficacy endpoints included the proportion of participants achieving the following at week 1 through week 12 and/or early discontinuation:

1. vIGA-AD of 0 or 1 with a ≥2-point improvement from baseline.
2. EASI-50, EASI-75, and EASI-90.
3. SCORAD-50, SCORAD-75, and SCORAD-90.

In addition, mean change from baseline in EASI and SCORAD was assessed at week 1 through week 12 and or/early discontinuation.

The statistical analyses assessing the efficacy of CD200R compared with the controls were conducted using the Fisher exact test for categorical outcomes and a mixed model for repeated measures for continuous outcomes that accounted for longitudinal data. The model used to calculate the $p$-values for both the percent change in SCORAD and EASI scores at 12 weeks included baseline, treatment, time point, and treatment-by-time point interaction. An unstructured covariance matrix was used to account for within-subject variability.

Clinical efficacy and safety analyses were performed using SAS® Version 9.4 or greater. Pharmacokinetic parameter estimates for ucenprubart in the clinical study were determined using noncompartmental procedures with Phoenix WinNonlin Version 8.1.1.

**Reporting summary**
Further information on research design is available in the Nature Portfolio Reporting Summary linked to this article.

## Data availability
The humanized mouse model scRNA-Seq data has been deposited under the GEO accession number GSE220685.

With respect to the phase 1 clinical study data, Lilly provides access to all individual participant data collected during the trial, after anonymization, with the exception of pharmacokinetic or genetic data. Data are available to request 6 months after the indication studied has been approved in the US and EU and after primary publication acceptance, whichever is later. No expiration date of data requests is currently set once data are made available. In order to ensure appropriate use and analysis of data, access is provided after a proposal has been approved by an independent review committee identified for this purpose and after receipt of a signed data sharing agreement and will be provided as soon as reasonably possible. Data and documents, including the study protocol, statistical analysis plan, clinical study report, and blank or annotated case report forms, will be provided in a secure data sharing environment. For details on submitting a request, see the instructions provided at www.vivli.org. All data are included in the Supplementary Information or available from the authors, as are unique reagents used in this Article. The raw numbers for charts and graphs are available in the Source Data file whenever possible. Source data are provided with this paper.

## Code availability
Readily available code from Seurat (v5.1.0) and SingleR (v.3.1.15) R packages was used for the analysis of scRNA-Seq data. Analysis scripts for Fig. 3 are available via a GitHub.

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

## Acknowledgements

This work is supported by Eli Lilly and Company. We thank Nada Alakhras, Colleen Burns, Gordafaried Deyanat-Yadzie, Beth Strifler, David Gemperline, and David Muench for technical assistance and execution of in vitro experiments. Thomas Melby, MS, of Syneos Health (Morrisville, NC, USA) provided writing assistance in the development of this paper, paid for by Eli Lilly and Company, and Jordan Bauer of Eli Lilly and Company provided assistance with data review. We thank the patients and subjects of these studies for their participation as well as the laboratory and clinical personnel who helped conduct them.

## Author contributions

Experiments were designed, executed, and/or analyzed by A.K., D.R.W., S.J.D., S.P., K.W., S.B., D.R., L.M., and A.G. The phase 1 clinical study was designed, executed, and/or analyzed by M.L., J.P., C.P., G.D., Z.W., J.K., D.M., D.P., C.S., P.K., and A.N. The paper was written by A.K., C.P., C.S., J.P. and A.N. with input from all authors.

## Competing interests

All authors with the exception of M.L. were employees, and many shareholders, of Eli Lilly and Company at the time of their initial work on this project, and declare no other competing interests. A.K., D.R.W., S.J.D., S.P., and D.R. hold a US patent (#11319370). M.L. is an investigator for AbbVie, Aclaris Therapeutics, Alfasigma, Almirall, Alumis, Anaptys-Bio, Arcutis Biotherapeutics, Arena Pharmaceuticals, Asana BioSciences, AstraZeneca, BELLUS Health, Boehringer Ingelheim, Brickell Biotech, Bristol Myers Squibb, Cara Therapeutics, Dermavant Sciences, Dermira, Eli Lilly and Company, Foamix Pharmaceuticals, Galderma, Incyte, Janssen, LEO Pharma, Pfizer, RAPT Therapeutics, Sun Pharma, and UCB.

## Additional information

[1]Lilly Biotechnology Center, San Diego, Eli Lilly and Company, Immunology Research, San Diego, CA, USA. [2]Lilly Research Laboratories, Eli Lilly and Company, Biotechnology, Indianapolis, IN, USA. [3]Progressive Clinical Research, San Antonio, TX, USA. [4]Lilly Biotechnology Center, San Diego, Eli Lilly and Company, Biotechnology, San Diego, CA, USA. [5]Lilly Research Laboratories, Eli Lilly and Company, Toxicology, Indianapolis, IN, USA. [6]Present address: Tentarix Biotherapeutics, San Diego, CA, USA. [7]Present address: Arcturus Therapeutics, San Diego, CA, USA. [8]Present address: Toralgen, Indianapolis, IN, USA. [9]Present address: Department of Biostatistics, Bloomberg School of Public Health, Johns Hopkins University, Baltimore, MD, USA. [10]Present address: Recludix Pharma, San Diego, CA, USA. ✉e-mail: anja.koster.phd@gmail.com

