## [Transparent Peer Review file · Nature Communications]

Ucenprubart is an agonistic antibody to CD200R with the potential to treat inflammatory skin disease: Preclinical development and a phase 1 clinical study

Corresponding Author: Dr Anja Koester

Version 1:

Reviewer comments:

Reviewer #1

(Remarks to the Author)

CD200R is a checkpoint inhibitor receptor recently shown to be upregulated in Th2 cells and in lesional atopic dermatitis skin, suggestive of a critical role in atopic dermatitis. The authors describe generation of a CD200R agonist IgG4 antibody (LY3454738), performed preclinical studies to demonstrate its ability to suppress in vitro ovalbumin-induced T cell activation and its lack of cytokine release through FcγR, and describe LY3454738 induced immune cell subpopulation transcriptional alterations distinct from dupilumab in a humanized mouse model of contact hypersensitivity. Phase 1 clinical results do not show serious adverse events for single ascending dose and repeat submaximal dosing and show modest efficacy with repeat dosing in atopic dermatitis patients. These results suggest that CD200R may potentially therapeutically ameliorate inflammatory/autoimmune disease and that LY3454738 has moderate potential to be a novel AD therapy.

Major critiques:

Please clarify experimental setup and results for Figure 1C. It does not appear to be described in methods and it's unclear what Figure 1c y-axis is measuring.

For experiments in figure 1, there appear to be no error bars or description of replicates, are these all single replicate experiments?

Figure 5B/C: Please define what % positive in dermis means on y-axis, e.g. for one GATA3 week 12 non-responder, the baseline lesion is 80%, presumably that doesn't mean 80% of dermal cells stain with GATA3? Also, what does the bottom of 5c refer to "(3/6 reduced from baseline)" under responders and "(8 of 13 reduced from baseline)" under nonresponders? Is this for CD3, GATA3, or the combination of the two? This and the results seem to suggest that for both responders and non-responders there is a similar trend towards a reduction in staining, which contradicts "CD3 and GATA3 staining were often evident in the dermis at study baseline but were lower at Week 12 following LY3454738 therapy (Fig. 5a and Fig. 5b); this trend was not observed in the EASI 75 nonresponders at Week 12." Related to this, in the methods for the IHC experiments, how GATA3 and CD3 positivity were quantitated does not appear to be described.

Minor critiques:

Line 87: Ideally, would describe relevance of cynomolgus activating receptor CD200RLa (cyCD200RLa) when first introduced (i.e. a closely related activating form of the Cd200R receptor) rather than later in the paragraph.

Lines 95-97: The experiment seems to show something slightly different than what is stated in results, that binding of LY3454738 is only moderately impacted by presence of CD200Fc ligand (i.e. as described in figure legend).

Line 98: Kd is presumably from surface plasmon resonance results described later on? (please clarify where this is from)

Line 98-99: The statement "demonstrating that the antibody does not target the ligand-binding site," is overly declarative as there are alternative explanations.

For Fig 1B and 1C, please fix triangles icons in figure (e.g. the triangles for ly3454738 are not universally pointing down) which is confusing.

Line 148: Please state assay/experiment done before describing QC steps (i.e. scRNA-seq of cutaneous immune cells was performed)

Line 233: It states "the percentage of positive cells in the dermis for GATA3 and CD3 233 demonstrated a strong correlation

(data not shown)", please clarify strong correlation to what?

Lines 592-598: For the Staphylococcal enterotoxin B T-cell proliferation assay methods, please revise to include addition of antibodies and at what concentration. Also, please clarify the role of abatacept as a control as its inclusion in the figure does not seem to be described anywhere in the paper.

Line 753: Please specify the epitope retrieval utilized.

Figure 5A: what are the insets on top left for GATA3 and CD3 staining? The text is too small to read.

Reviewer #2

(Remarks to the Author)

Comments to authors

Overall, the statistical analyses are appropriate for the design and the hypotheses being tested. For example, the use of repeated measures models for comparing within- and between-group changes from baseline is perfectly appropriate.

However, there are a number of minor concerns with the presentation and description of the design and analyses.

1. Figure 3b. The axes are labeled 'UMAP-1' and 'UMAP-2' with ranges that run from -15 to 10 and from -20 to 20. What do these axes represent? How would the reader interpret these figures, other than to conclude that the clustering produces expressions that can be linked to known cell types? IS there anything to be concluded from the cells that are clustered away from their main cluster?

2. Figure 3d. It is not clear to me that the Wilcoxon test is appropriate here, since the differentially expressed genes are likely correlated with one another. Having said that, the differences in Figure 3d are so large that a statistical tests is not really needed.

3. Figure 2. The samples appear to be from 6 donors and are treated with LY3454738 IgG4 S228P (IgG4P), LY3454738 IgG4P F234A/L235A (IgG4PAA), etc. Technically, the analysis should take this into account, though again, the differences appear to be sufficiently robust that this correction is unlikely to have an effect on the results.

4. The protocol was not included in the supplemental material. There was a CONSORT diagram but no description of the dose-escalation design. That leaves a number of unanswered questions about the dose-finding design. How were dose escalations decided on? Why was the 500mg dose chosen for study c? What were the TEAEs for each dose? This information could be put into supplemental material.

Reviewer #3

(Remarks to the Author)

The manuscript by Koester et al. presents new data on the development, in vitro testing, in vivo testing and a Phase I clinical trial of an agonistic antibody for the treatment of atopic dermatitis, a common chronic inflammatory skin disease. The approach is novel, promising and therefore of high interest.

The data on development of the antibody LY3454738 are well-described, but it is sometimes not clear, how many replicates were performed. The in vivo testing in a humanized mouse model of delayed-time hypersensitivity to a hapten was followed by single-cell sequencing of cells infiltrating the skin. The application to patients had apparently beneficial effects, although these are not depicted clearly in the Figure. Finally the authors present data from histological stainings of AD lesions before and after therapy, which are too small to show an effect. To build a bridge between the findings, it would be beneficial to perform further in vitro experiments that reflect directly the hypothesis (e.g., CD200R-expressing macrophages and their ability to attract GATA3+ T cells). The study is written in a coherent and understandable manner and in proper English. The data are promising, but is some parts preliminary.

I have the following concerns:

Figure Legend 1 needs a statement on how many replicates were performed. Figure 1d shows normalized data. It would be needed to show how strong the initial IL-2 response was in the ELISA to convince the readers.

Figure Legend 2 needs a statement on how many replicates were performed. How many healthy donors were included for the experiments shown in Figure 1c?

It would be interesting to see if the treatment with LY3454738 did change the expression levels of CD200R when compared to placebo or Dupilumab.

It appears as if roughly same numbers of monocytes and macrophages were received from the mouse ears in Figure 3B, although in tissue, rather macrophages are to be expected. Could the authors comment on this.

The statistically significant differences from placebo, which are described in the text, should also be made visible in the figure 4b and 4e.

The histological data does unfortunately not show significant differences. It should be stated more clearly how large the area was, that was used for GATA3+ and CD3+ cell quantification, and how it was defined. What do the % refer to, is it stained dermal nuclei? It may make more sense to count the GATA3+ among CD3+. If biopsy material is still available, this would add significant value to the measurement. The biopsies from placebo control donors should be included in this analysis as well. Although images are clear, isotype control stains would be preferred. However, if there are no differences, the results would need to be shifted into the supplement.

As stated above, the study would be more compelling if it could be shown that the cells most affected by LY3454738, macrophages, are involved in Th2 recruitment into the skin and that LY interferes with this.

Version 2:

Reviewer comments:

Reviewer #1

(Remarks to the Author)

Thanks for addressing previous concerns. No further issues.

Reviewer #2

(Remarks to the Author)

Thank you for your detailed and thoughtful responses to my comments. My main concerns on the initial submission were 1) lack of details on the design and 2) statistical analyses that did not account for within-donor correlations. For the first comment, the authors provided the protocol which described the single dose escalation study "design". The allocation of doses to participants had a formal structure (no escalation if 2 or more TEAEs, etc.) but it also allowed for so many ad hoc decisions that the properties of the design are hard to determine. But that is unfortunately how most early phase trials are done.

For the second comment, the authors refer to a previous paper that did a "one-way ANOVA followed by a paired t-test." The previous paper does not justify this approach either. The one-way ANOVA does not account for the within-donor correlations and so technically, is not the proper analysis in either the previous or current paper. I am wondering if the authors are using 'pairwise comparisons' and 'paired comparisons' interchangeably? From a statistical point of view, they are fundamentally different. However, as noted in the initial review, the effects are so large that the proper analyses would almost surely yield the same results.

Reviewer #3

(Remarks to the Author)

The manuscript has evolved significantly and is much easier to understand.

The authors might want to correct in Figure Legend 6 the assigning of b and c to CD3 and GATA3, which appear to be mixed up.

The manuscript would attract more interest, if the DEGs of figure 3f (especially those that lead to fig. 3g) would be shown in a supplemental table.

REVIEWER COMMENTS

Reviewer #1 (Remarks to the Author):

CD200R is a checkpoint inhibitor receptor recently shown to be upregulated in Th2 cells and in lesional atopic dermatitis skin, suggestive of a critical role in atopic dermatitis. The authors describe generation of a CD200R agonist IgG4 antibody (LY3454738), performed preclinical studies to demonstrate its ability to suppress in vitro ovalbumin-induced T cell activation and its lack of cytokine release through FcγR, and describe LY3454738 induced immune cell subpopulation transcriptional alterations distinct from dupilumab in a humanized mouse model of contact hypersensitivity. Phase 1 clinical results do not show serious adverse events for single ascending dose and repeat submaximal dosing and show modest efficacy with repeat dosing in atopic dermatitis patients. These results suggest that CD200R may potentially therapeutically ameliorate inflammatory/autoimmune disease and that LY3454738 has moderate potential to be a novel AD therapy.

REVIEWER COMMENT

Major critiques:

1) Please clarify experimental setup and results for Figure 1C. It does not appear to be described in methods and it's unclear what Figure 1c y-axis is measuring.

AUTHORS' RESPONSE

Thank you for pointing out the omission of the methods for this experiment. We have added the experimental procedure to the methods, changed the y-axis label, and updated the legend, including adding an explanation of the scaling of the y-axis. We also decided to switch panels 1c and 1b in Fig. 1. The legend for Fig.1b (old Fig. 1c) now reads as follows:

“Flow cytometry showed ucenprubart binding to human (hCD200R) and cynomolgus (cyCD200R) CD200R but not to cyCD200RLa in CHO cells recombinantly expressing the respective receptors. Note that ucenprubart has higher affinity to cynomolgus (triangles) versus human (squares) receptor; as a result, the binding signal is higher for cynomolgus receptor-expressing cells. No binding was detected on cells expressing cyCD200RLa and nontransfected cells (downward triangles and circles, respectively; overlapping along x-axis) Values are the average of 2 technical repeats ± SEM.”

REVIEWER COMMENT

2) For experiments in figure 1, there appear to be no error bars or description of replicates, are these all single replicate experiments?

AUTHORS' RESPONSE

Thank you for the question. We have replaced Fig. 1c (old Fig. 1b) with new data and a graph that we believe better demonstrates the ability of the ligand to bind to the receptor in the presence of the antibody. We have updated the text (~lines 124-129) in the revision showing

track changes, as shown below) and the figure legend. The experiment now shown was conducted with duplicates and the panel has error bars (standard error of the mean).

“The ability of the ligand to bind cell-bound CD200R in the presence of ucenprubart was further confirmed in a flow competition experiment using HEL92.1.7 cells naturally expressing CD200R (Fig. 1c). Cells were incubated with ucenprubart, CD200Fc, or isotype antibody followed by a titration of the ligand. The binding of the CD200Fc ligand increased slightly when ucenprubart was prebound to the cells, demonstrating that the antibody does not target the ligand-binding site.”

Fig. 1c: We have replaced the previous graph with a new figure containing duplicates and error bars for each data point and edited the figure legend accordingly.

Fig. 1d shows normalized data. This was necessary because compounds were tested in different assays over the span of 2 years and had to be normalized for comparison. IL-2 response to ovalbumin varied between experiments from 800 to 1200 pg/mL compared to background of less than 5 pg/mL. All molecules were tested in duplicates; for normalization, the average was converted to percentage of baseline. We have changed the figure legend to add the following explanation:

“Values shown are compiled from multiple experiments and normalized to percentage of baseline for comparison. Actual IL-2 values after stimulation with ovalbumin peptide varied between 800 to 1200 pg/mL compared to <5 pg/mL unstimulated.”

REVIEWER COMMENT

3) Figure 5B/C: Please define what % positive in dermis means on y-axis, e.g. for one GATA3 week 12 non-responder, the baseline lesion is 80%, presumably that doesn't mean 80% of dermal cells stain with GATA3?

AUTHORS' RESPONSE

Thank you for your question. You are correct with your assertion. The region of interest was drawn below the epidermis containing all the cells within the dermis, except for those within adnexal structures (hair follicles, sebaceous glands, sweat glands, etc.). Image analysis was performed for CD3 staining, while 2 pathologists reviewed the areas to score percent positivity for GATA3 staining. We have updated the y-axis of the graph to “% Positive Immune Cells / All Cells in the Dermis” for clarity. We have added the following sentence to the Methods section (~lines 704-706):

“For CD3, image analysis was performed on the dermis, excluding the epidermis and adnexal structures of the skin. GATA3 positivity was determined on the same region of interest by 2 pathologists scoring the percentage of positive immune cells among all cells in the dermis.”

REVIEWER COMMENT

4) Also, what does the bottom of 5c refer to “(3/6 reduced from baseline)” under responders and “(8 of 13 reduced from baseline)” under nonresponders? Is this for CD3, GATA3, or the

combination of the two? This and the results seem to suggest that for both responders and non-responders there is a similar trend towards a reduction in staining, which contradicts “CD3 and GATA3 staining were often evident in the dermis at study baseline but were lower at Week 12 following LY3454738 therapy (Fig. 5a and Fig. 5b); this trend was not observed in the EASI 75 nonresponders at Week 12.”

AUTHORS' RESPONSE

Thank you for your thorough review and for identifying this issue. Firstly, there was a missing statement under Fig. 5b, where the words “(5/6 reduced from baseline) and (7/13 reduced from baseline)” should have been shown. This omission certainly made the figure more confusing. Secondly, we agree with your assessment that our writing was unclear and our conclusions lacking. While we do not see any statistically significant differences between the responder and nonresponder groups, due to the low numbers of responders, we do see numerical trends within the GATA3 staining that we feel help with hypothesis generation (5/6 reduced in the responder group). We have therefore calculated the delta difference between baseline staining and week 12 values for both CD3 and GATA3 and added those data to the figure as new panels. These new panels show that there is a larger mean reduction in GATA3 in the responders compared to the nonresponders. Because there is no difference for CD3 staining, we have amended the text to the following (starting at line 268 in the revision showing track changes):

“In the dermis of lesional punch biopsies from patients with AD at baseline and week 12, the percentage of cells staining for GATA3 was positively correlated with cells staining for CD3 (Pearson $r = 0.7546$, $p < 0.0001$; data not shown), suggesting the presence of Th2 cells within this compartment. For CD3, within-patient differences were not observed between baseline and week 12 lesional samples for either EASI-75 responders or nonresponders. For GATA3, 5 of 6 EASI-75 responders showed reduced GATA3 staining in the dermis between baseline and week 12 (Fig. 5a and 5b); this trend was not observed for GATA3 among EASI-75 nonresponders (Fig. 5c). When the mean change between baseline and week 12 was calculated, the ucenprubart responder group had a numerically greater reduction in GATA3-positive staining compared with nonresponders (Fig. 5c). CD3 is a general marker of T cells in the skin while GATA3 more specifically identifies Th2 cells in the dermis, suggesting that pathogenic T cells in ucenprubart responders tended to be reduced when compared to nonresponders.”

REVIEWER COMMENT

5) Related to this, in the methods for the IHC experiments, how GATA3 and CD3 positivity were quantitated does not appear to be described.

AUTHORS' RESPONSE

Thank you for the critique of the methods, as we were not clear. We have added the following to the Methods section (~lines 704-706, track changes version) to improve the transparency of our quantitation methods.

“For CD3, image analysis was performed on the dermis, excluding the epidermis and adnexal structures of the skin. GATA3 positivity was determined on the same region of interest by 2 pathologists scoring the percentage of positive immune cells among all cells in the dermis.”

REVIEWER COMMENT

Minor critiques:

Line 87: Ideally, would describe relevance of cynomolgus activating receptor CD200RLa (cyCD200RLa) when first introduced (i.e. a closely related activating form of the Cd200R receptor) rather than later in the paragraph.

AUTHORS' RESPONSE

Thank you for pointing out the poor description; we have added a statement about the activating cynomolgus receptor. The text now reads as follows (~lines 103-107):

“CD200R is a member of the paired receptor family, meaning a closely related homologue of the inhibitory CD200R with opposite activity exists in cynomolgus monkeys (cyCD200RLa). To enable non-human primate toxicological studies, we counter screened by flow cytometry for non-binding on cells expressing cyCD200RLa.”

Lines 95-97: The experiment seems to show something slightly different than what is stated in results, that binding of LY3454738 is only moderately impacted by presence of CD200Fc ligand (i.e. as described in figure legend).

AUTHORS' RESPONSE

Thank you for the comment and identifying this error. We have repeated the experiment in a different set up in which the cells were incubated with the high affinity antibody first and then titrated the ligand with the lower affinity, which demonstrates that the ligand binding is slightly enhanced in the presence of the antibody. We believe this can be explained by a conformational change induced by the antibody that facilitates ligand binding. We have changed the figure, the corresponding figure legend, and the text. We have edited the text as follows (~lines 124–129):

“The ability of the ligand to bind cell-bound CD200R in the presence of ucenprubart was further confirmed in a flow competition experiment using HEL92.1.7 cells naturally expressing CD200R (Fig. 1c). Cells were incubated with ucenprubart, CD200Fc, or isotype antibody followed by a titration of the ligand. The binding of the CD200Fc ligand increased slightly when ucenprubart was prebound to the cells, demonstrating that the antibody does not target the ligand-binding site.”

Line 98: Kd is presumably from surface plasmon resonance results described later on? (please clarify where this is from)

AUTHORS' RESPONSE

We have edited this section to read as follows (~lines 124-129):

“The ability of the ligand to bind cell-bound CD200R in the presence of ucenprubart was further confirmed in a flow competition experiment using HEL92.1.7 cells naturally expressing CD200R (Fig. 1c). Cells were incubated with ucenprubart, CD200Fc, or isotype antibody followed by a titration of the ligand. The binding of the CD200Fc ligand increased slightly when ucenprubart was prebound to the cells, demonstrating that the antibody does not target the ligand-binding site.”

REVIEWER COMMENT

Line 98-99: The statement “demonstrating that the antibody does not target the ligand-binding site,” is overly declarative as there are alternative explanations.

AUTHORS’ RESPONSE:

We believe that with the updated experiment and data this statement is now acceptable.

REVIEWER COMMENT

For Fig 1B and 1C, please fix triangles icons in figure (e.g. the triangles for ly3454738 are not universally pointing down) which is confusing.

AUTHORS’ RESPONSE

Thank you for noticing this inconsistency. We have updated the graphs to be consistent. Note that Fig. 1b and Fig. 1c were switched for better flow.

REVIEWER COMMENT

Line148: Please state assay/experiment done before describing QC steps (i.e. scRNA-seq of cutaneous immune cells was performed)

AUTHORS’ RESPONSE

Thank you for pointing out this omission. We have added a brief description of the experiment to the text as follows (~lines 161-166):

“CD34-engrafted NSG mice (huNOG-EXL) were sensitized to the hapten oxazolone and dosed with either ucenprubart or dupilumab 24 hours before challenge with hapten on the ear. Efficacy was determined by measuring ear thickness before and after the challenge and compared to isotype control antibody-treated animals. The challenge was repeated 3 times. Following the third challenge (day 19 post-sensitization), dupilumab and ucenprubart each reduced swelling by approximately 50% during the 24-hour challenge (Fig. 3a).”

REVIEWER COMMENT

Line 233: It states “the percentage of positive cells in the dermis for GATA3 and CD3 demonstrated a strong correlation (data not shown)”, please clarify strong correlation to what?

AUTHORS’ RESPONSE

Thank you for the question. We have clarified the poorly worded statement to the following to indicate that GATA3 and CD3 were positively correlated:

“In the dermis of lesional punch biopsies from patients with AD at baseline and week 12, the percentage of cells staining for GATA3 was positively correlated with cells staining for CD3 (Pearson $r = 0.7546$, $p < 0.0001$; data not shown), suggesting the presence of Th2 cells within this compartment.”

REVIEWER COMMENT

Lines 592-598: For the Staphylococcal enterotoxin B T-cell proliferation assay methods, please revise to include addition of antibodies and at what concentration. Also, please clarify the role of abatacept as a control as its inclusion in the figure does not seem to be described anywhere in the paper.

AUTHORS' RESPONSE

Thank you for pointing out the missing information. We have revised the legend of Fig. 2c to include the antibody concentration and a statement about the purpose of abatacept as a positive control for an inhibitor of T-cell activation as follows:

“Antibodies were added at 30 $\mu\text{g}/\text{mL}$ immediately before stimulation for 96 hours. Data shown are triplicates from a total of 4 donors. Abatacept (CTLA4-Fc) was used a positive control for inhibition of T-cell activation.”

REVIEWER COMMENT

Line 753: Please specify the epitope retrieval utilized.

AUTHORS' RESPONSE

Thank you for your interest in the immunohistochemistry methodology. We utilized BOND Epitope Retrieval Solution 2 from Leica. We have updated the text to the following (starting at ~line 699)

“For immunohistochemistry, specifically, tissue sections were pretreated with BOND Epitope Retrieval Solution 2 (Leica, catalog no. AR9640) before the addition of mAbs against CD3 (clone LN10; Leica) and GATA3 (clone L50-823; Abcam) using the Leica BOND RX autostainer with CDX-capable protocols with red chromogen.”

REVIEWER COMMENT

Figure 5A: what are the insets on top left for GATA3 and CD3 staining? The text is too small to read.

AUTHORS' RESPONSE

We appreciate your attention to detail on the figures. We have enlarged the immunohistochemistry images to improve visibility to the reader, but the insets in the top left of the GATA3 and CD3 staining are coded labels from the samples themselves and cannot be

removed from the image. We hope that by enlarging the image, the reader will be able to better see the changes in the dermis and not be distracted by the label in the corner.

Reviewer #2 (Remarks to the Author):

Comments to authors

Overall, the statistical analyses are appropriate for the design and the hypotheses being tested. For example, the use of repeated measures models for comparing within- and between-group changes from baseline is perfectly appropriate. However, there are a number of minor concerns with the presentation and description of the design and analyses.

REVIEWER COMMENT

1) Figure 3b. The axes are labeled 'UMAP-1' and 'UMAP-2' with ranges that run from -15 to 10 and from -20 to 20. What do these axes represent? How would the reader interpret these figures, other than to conclude that the clustering produces expressions that can be linked to known cell types? IS there anything to be concluded from the cells that are clustered away from their main cluster?

AUTHORS' RESPONSE

We thank the reviewer for this comment and would like to clarify the labels. The label axes in Fig. 3b stand for Uniform Manifold Approximation and Projection (UMAP). UMAP is a widely used nonlinear dimension reduction technique for single-cell data to visualize patterns of cell clustering in a 2-dimensional space. As pointed out by the reviewer, the reader can conclude the presence of major cell types in the single-cell data set represented by the UMAP approach and identify subsets across cells clusters (eg, CD4 vs CD8 T cells in Fig. 3b). The UMAP axes are arbitrary and separate the single-cell data projected to a low dimensional UMAP space. There is no intuitive interpretation for the distances in UMAP projections. Cells that are clustered separately from their main cluster tend to have dissimilar expression signatures. Cells within each cluster are likely to share more similar expression signatures and belong to the same cell type.

REVIEWER COMMENT

2) Figure 3d. It is not clear to me that the Wilcoxon test is appropriate here, since the differentially expressed genes are likely correlated with one another. Having said that, the differences in Figure 3d are so large that a statistical test is not really needed.

AUTHORS' RESPONSE

We agree on the large differences that were observed with treatment for a subset of cells. The nonparametric Wilcoxon rank-sum test is a commonly used method to identify differentially expressed genes across conditions for different cell types in single-cell data (Squair et al., 2021). It is suited for smaller single-cell data sets such as ours that lack the number of replicates to perform pseudo-bulk analysis. However, the Wilcoxon rank-sum test can increase the risk of false discoveries. For this reason, we have outlined the limitations of our single gene test in the Discussion section of our manuscript (paragraph starting with line 357) and performed pathway analysis to obtain more robust results beyond the single gene level.

Reference: Squair, J. W. et al. Confronting false discoveries in single-cell differential expression. *Nat Commun* **12**, 5692 (2021).

REVIEWER COMMENT

3) Figure 2. The samples appear to be from 6 donors and are treated with LY3454738 IgG4 S228P (IgG4P), LY3454738 IgG4P F234A/L235A (IgG4PAA), etc. *Technically, the analysis should take this into account, though again,* the differences appear to be sufficiently robust that this correction is unlikely to have an effect on the results.

AUTHORS' RESPONSE

We appreciate the scrutiny of the statistical analysis. If we understand correctly, the reviewer is suggesting we should use a paired test to account for the fact that the same donors are tested across all treatments. If so, this has already been considered as the p values were generated by 1-way ANOVA and followed by a paired t test (this is stated in the figure legend). Reference 21 (Alakhras et al., 2018) describes in more detail the statistical approach used.

REVIEWER COMMENT

4) The protocol was not included in the supplemental material. There was a CONSORT diagram but no description of the dose-escalation design. That leaves a number of unanswered questions about the dose-finding design. How were dose escalations decided on? Why was the 500mg dose chosen for study c? What were the TEAEs for each dose? This information could be put into supplemental material.

AUTHORS' RESPONSE

We apologize for the oversight regarding the inclusion of the protocol with the submission, which was due to misunderstanding where to include it in the submission. We have included the protocol with the submission of our response. In addition, we added to the Supplementary Materials receptor occupancy data from the ascending dose portion of the clinical study to support the dose selection for Part C.

Reviewer #3 (Remarks to the Author):

The manuscript by Koester et al. presents new data on the development, in vitro testing, in vivo testing and a Phase I clinical trial of an agonistic antibody for the treatment of atopic dermatitis, a common chronic inflammatory skin disease. The approach is novel, promising and therefore of high interest.

The data on development of the antibody LY3454738 are well-described, but it is sometimes not clear, how many replicates were performed. The in vivo testing in a humanized mouse model of delayed-time hypersensitivity to a hapten was followed by single-cell sequencing of cells infiltrating the skin. The application to patients had apparently beneficial effects, although these are not depicted clearly in the Figure.

REVIEWER COMMENT

Finally the authors present data from histological stainings of AD lesions before and after therapy, which are too small to show an effect.

AUTHORS' RESPONSE

Thank you for the feedback. We agree they are difficult to see, so we have increased the size of the figure to make the images easier to view and added panels showing mean values for staining at baseline and week 12.

REVIEWER COMMENT

To build a bridge between the findings, it would be beneficial to perform further in vitro experiments that reflect directly the hypothesis (e.g., CD200R-expressing macrophages and their ability to attract GATA3+ T cells). The study is written in a coherent and understandable manner and in proper English. The data are promising, but in some parts preliminary.

AUTHORS' RESPONSE

The authors agree that we only have preliminary data to support the GATA3 results. We agree that your proposed experiment would make for a more robust analysis. Unfortunately, we do not have the ability to perform additional experiments. Because our immunohistochemistry results are more for hypothesis generation rather than confirmatory in nature, we have softened our language and conclusions.

I have the following concerns:

REVIEWER COMMENT

Figure Legend 1 needs a statement on how many replicates were performed. Figure 1d shows normalized data. It would be needed to show how strong the initial IL-2 response was in the ELISA to convince the readers.

AUTHORS' RESPONSE

Thank you for your comment. We have updated the figures and included replicates where possible (Fig. 1b and Fig. 1c). The reviewer is correct that Fig. 1d is normalized data. Compounds were tested in different assays and had to be normalized for comparison. IL-2 response to ovalbumin peptide varied between experiments from 800 to 1200 pg/mL compared to background readings of less than 5 pg/mL. All molecules were tested in duplicate; for normalization, the average was converted to percentage of baseline. We have changed the legend for Fig. 1d and added an explanation as follows:

“In ovalbumin-activated T cells co-cultured with CD200-expressing B cells, IL-2 release was inhibited with ucenprubart but was increased by an anti-CD200 antibody. Values shown are compiled from multiple experiments and normalized to percentage of baseline for comparison. Actual IL-2 values after stimulation with ovalbumin peptide varied from 800 to 1200 pg/mL compared to <5 pg/mL unstimulated.”

REVIEWER COMMENT

Figure Legend 2 needs a statement on how many replicates were performed. How many healthy donors were included for the experiments shown in Figure 1c?

AUTHORS' RESPONSE

We added the number of technical replicates for Fig. 2b in the legend as follows:

“Values are the average of 2 technical replicates \pm SEM.”

We added the number of donors tested in the experiment depicted in the Fig. 2c legend as follows:

“Data shown are triplicates from a total of 4 donors.”

REVIEWER COMMENT

It would be interesting to see if the treatment with LY3454738 did change the expression levels of CD200R when compared to placebo or Dupilumab.

AUTHORS' RESPONSE

We agree with the reviewer's observation and have added a plot of CD200R CITE-seq surface marker expression in the Supplementary Materials. We had previously found that CD200R1 gene expression levels (RNA) were below detectable levels across cell types, which hampered our ability to compare ucenprubart- to dupilumab-treated cells. We did not see significant changes in CD200R surface marker expression across cell types when comparing isotype controls to ucenprubart- or dupilumab-treated ear skin samples from the oxazolone mouse model (Supplementary Fig. 2).

REVIEWER COMMENT

It appears as if roughly same numbers of monocytes and macrophages were received from the mouse ears in Figure 3B, although in tissue, rather macrophages are to be expected. Could the authors comment on this.

AUTHORS' RESPONSE

We thank the reviewer for their helpful insights related to the proportions of infiltrating myeloid cells in the ears of our oxazolone mouse model. Our results are in line with a published single-cell study that identified a similar proportion of myeloid and T cells in the ears of oxazolone-treated mice when compared to nontreated controls (Liu et al., 2020). This study also found a similar proportion of monocytes and macrophages in tissue from oxazolone-treated skin. The authors highlighted that a subset of the identified monocytes expressed both macrophage and monocyte markers and classified them as a monocyte/monocyte-derived macrophage subset. Differentiating between these heterogenous monocyte and macrophage populations in our data set is challenging based on single-cell RNA-Seq markers alone and the limited number of CITE-seq surface markers available. Therefore, we cannot exclude the possibility that a subset of the observed monocytes in our data set include “monocyte-derived macrophages” expressing both macrophage and monocyte markers. While it will be interesting to characterize these monocyte subsets in greater detail, we wanted to focus on the differences of the major cell types in the current study to better understand differences in response to ucenprubart and dupilumab.

Reference: Liu, Y. et al. Single-cell profiling reveals divergent, globally patterned immune responses in murine skin inflammation. *iScience* **23**, 101582 (2020).

REVIEWER COMMENT

The statistically significant differences from placebo, which are described in the text, should also be made visible in the figure 4b and 4e.

AUTHORS' RESPONSE

The statistically significant differences from placebo were values for the overall change between ucenprubart and placebo across the 12-week period; as such, they are not easily visualized. We have added the values to the legends of Fig. 4b and 4d, mirroring the text in the Results section, as follows:

Fig. 4b: “(overall mean percentage change in EASI score between ucenprubart and placebo, $p = 0.0157$).”

Fig 4d: “(overall mean percentage change in SCORAD between ucenprubart and placebo, $p = 0.004$).”

REVIEWER COMMENT

The histological data does unfortunately not show significant differences. It should be stated more clearly how large the area was, that was used for GATA3+ and CD3+ cell quantification, and how it was defined. What do the % refer to, is it stained dermal nuclei? It may make more

sense to count the GATA3+ among CD3+. If biopsy material is still available, this would add significant value to the measurement

AUTHORS' RESPONSE

Thank you for your insightful critique. We agree that the method of quantification was not clear in the methods. We have added the following to the Methods section (~lines 703-706) to improve the transparency of our quantitation methods:

“For CD3, image analysis was performed on the dermis, excluding the epidermis and adnexal structures of the skin. GATA3 positivity was determined on the same region of interest by 2 pathologists scoring the percentage of positive immune cells among all cells in the dermis.”

Additionally, we agree that the experiment you propose would be optimal. Unfortunately, we are unable to perform such experiments due to logistical challenges. Our desire was to present these immunohistochemistry results primarily as background for hypothesis generation. The experiment you propose could be performed in a more thorough mechanism of analysis study in larger patient cohorts where we would have more power to answer these important questions.

REVIEWER COMMENT

The biopsies from placebo control donors should be included in this analysis as well.

AUTHORS' RESPONSE

Thank you for the suggestion. We did not include placebo control samples in the manuscript, instead opting for baseline and week 12 results from the same subject. We had very few placebo responders to perform an appropriate analysis.

REVIEWER COMMENT

Although images are clear, isotype control stains would be preferred. However, if there are no differences, the results would need to be shifted into the supplement.

AUTHORS' RESPONSE

We appreciate your suggestion to include isotype control stains for the immunohistochemistry measurements. These methods were validated at a contract research laboratory, with controls included during validation. Further use of isotype controls was not utilized after the methods were fully developed.

REVIEWER COMMENT

As stated above, the study would be more compelling if it could be shown that the cells most affected by LY3454738, macrophages, are involved in Th2 recruitment into the skin and that LY interferes with this.

AUTHORS' RESPONSE

The authors agree with the reviewer that an in vitro experiment directly demonstrating the inhibition of Th2 cells by ucnprubart-treated macrophages would be an elegant experiment and excellent proof of the proposed mechanism of action. However, given the already very broad scope of the manuscript, we plan to explore this hypothesis in a future manuscript.

REVIEWERS' COMMENTS

Reviewer #1 (Remarks to the Author):

Thanks for addressing previous concerns. No further issues.

Reviewer #2 (Remarks to the Author):

Thank you for your detailed and thoughtful responses to my comments. My main concerns on the initial submission were 1) lack of details on the design and 2) statistical analyses that did not account for within-donor correlations. For the first comment, the authors provided the protocol which described the single dose escalation study "design". The allocation of doses to participants had a formal structure (no escalation if 2 or more TEAEs, etc.) but it also allowed for so many ad hoc decisions that the properties of the design are hard to determine. But that is unfortunately how most early phase trials are done.

For the second comment, the authors refer to a previous paper that did a "one-way ANOVA followed by a paired t-test." The previous paper does not justify this approach either. The one-way ANOVA does not account for the within-donor correlations and so technically, is not the proper analysis in either the previous or current paper. I am wondering if the authors are using 'pairwise comparisons' and 'paired comparisons' interchangeably? From a statistical point of view, they are fundamentally different. However, as noted in the initial review, the effects are so large that the proper analyses would almost surely yield the same results.

Thanks for pointing out this potential source of misunderstanding.

A paired t-test is used to compare the treatments. Please excuse the unclear language in the description; we have changed the figure legend accordingly.

Reviewer #3 (Remarks to the Author):

The manuscript has evolved significantly and is much easier to understand.

The authors might want to correct in Figure Legend 6 the assigning of b and c to CD3 and GATA3, which appear to be mixed up.

Thank you – when we updated the bar graphs to show individual data points, we corrected this.

The manuscript would attract more interest, if the DEGs [differentially expressed genes] of figure 3f (especially those that lead to fig. 3g) would be shown in a supplemental table.

Provided as Supplementary Table 3.